# Cu Homeostasis in Bacteria: The Ins and Outs

**DOI:** 10.3390/membranes10090242

**Published:** 2020-09-18

**Authors:** Andreea Andrei, Yavuz Öztürk, Bahia Khalfaoui-Hassani, Juna Rauch, Dorian Marckmann, Petru-Iulian Trasnea, Fevzi Daldal, Hans-Georg Koch

**Affiliations:** 1Institut für Biochemie und Molekularbiologie, ZBMZ, Medizinische Fakultät, Albert-Ludwigs Universität Freiburg; Stefan Meier Str. 17, 79104 Freiburg, Germany; Andreea.Andrei@biochemie.uni-freiburg.de (A.A.); yavuz.oz48@outlook.com (Y.O.); juna.rauch@students.uni-freiburg.de (J.R.); dorian.marckmann@gmx.de (D.M.); 2Fakultät für Biologie, Albert-Ludwigs Universität Freiburg; Schänzlestrasse 1, 79104 Freiburg, Germany; 3Université de Pau et des Pays de l’Adour, CNRS, IPREM UMR 5254, BP1155 Pau, France; b.khalfaoui-hassani@univ-pau.fr; 4Institute of Science and Technology, Am Campus 1, 3400 Klosterneuburg, Austria; petruiulian.trasnea@ist.ac.at; 5Department of Biology, University of Pennsylvania, Philadelphia, PA 19104, USA; fdaldal@sas.upenn.edu

**Keywords:** P_1B_-type ATPases, copper chaperones, major facilitator superfamily proteins, cupric reductase, metal-sensing transcriptional regulators, *cbb*_3_-type cytochrome *c* oxidase

## Abstract

Copper (Cu) is an essential trace element for all living organisms and used as cofactor in key enzymes of important biological processes, such as aerobic respiration or superoxide dismutation. However, due to its toxicity, cells have developed elaborate mechanisms for Cu homeostasis, which balance Cu supply for cuproprotein biogenesis with the need to remove excess Cu. This review summarizes our current knowledge on bacterial Cu homeostasis with a focus on Gram-negative bacteria and describes the multiple strategies that bacteria use for uptake, storage and export of Cu. We furthermore describe general mechanistic principles that aid the bacterial response to toxic Cu concentrations and illustrate dedicated Cu relay systems that facilitate Cu delivery for cuproenzyme biogenesis. Progress in understanding how bacteria avoid Cu poisoning while maintaining a certain Cu quota for cell proliferation is of particular importance for microbial pathogens because Cu is utilized by the host immune system for attenuating pathogen survival in host cells.

## 1. Introduction

Approximately 30% of all proteins in bacteria depend on metals for their function. Understanding how these potentially toxic metals are imported into bacterial cells and how they are ultimately delivered to their target proteins without inducing toxic effects is a crucial issue in metalloprotein biogenesis (Figure 1) [1,2]. In contrast to most other nutrients, the concentration of many metals in natural environments usually exceeds cellular needs and hence multiple mechanisms that prevent metal-induced damage are encountered in bacteria. Copper (Cu) is one such metal that is essential in eukaryotes and prokaryotes but is highly toxic when present in excess. Indeed, Cu alloys are used for controlling bacterial surface contaminations [3]. Furthermore, human macrophages pump copper into their phagosomes after engulfing pathogenic bacteria to induce oxidative stress and bacterial cell death [4]. Consequently, bacteria have to deal with high copper concentrations in order to evade the host immune system [4,5,6,7]. The importance of Cu detoxifying systems for bacterial virulence is exemplified by the observation that the inactivation of Cu-exporting P_1B_-type ATPases in *Mycobacterium tuberculosis* impairs their ability to proliferate in host macrophages [8,9]. The link between Cu homeostasis and bacterial virulence is summarized in several recent reviews and not covered in depth here [5,10,11,12,13,14,15].

Cu toxicity is intrinsically linked to its redox properties that favor the generation of reactive oxygen species via a Fenton-like reaction:Cu^+^ + H_2_O_2_ → Cu^2+^ + OH^−^ + ^·^HO

In addition, Cu is located on top of the Irving-Williams series and Cu binding to proteins is usually a thermodynamically favored process [16]. Although this aids Cu insertion into cuproenzymes, excess Cu could lead to significant mis-metalation of proteins naturally containing other metals, and iron in particular, resulting in their inactivation [17,18,19]. Potent targets of Cu toxicity in the periplasmic space of bacteria are the biogenesis pathways for cytochrome *c* [19] and bacteriochlorophyll [20], aside from interfering with the thiols of periplasmic proteins such as the thiol:disulfide oxidoreductases [21].

The high redox potential of the Cu(II)/Cu(I) pair (+160 mV) favors reactions with oxygen and oxygen containing molecules and it is generally assumed that Cu-dependent proteins have evolved concomitantly with the appearance of molecular oxygen, which started about 3 × 10^9^ years ago [22]. Studies on Cu transport have mainly focused on Cu export for Cu detoxification, in line with its harmful effects, while the mechanisms of Cu import across the outer membrane of Gram-negative bacteria and the bacterial cytoplasmic membrane have been analyzed in only a few cases (Figure 1) [23,24].

Cu-binding proteins (cuproproteins) in the bacterial cytosol generally bind Cu only transiently and act mainly as Cu-chaperones, Cu storage proteins and Cu-responsive transcriptional regulators [25]. In contrast, cuproenzymes, which contain Cu as part of their catalytic center and bind Cu permanently, are primarily involved in aerobic and anaerobic electron transfer reactions, monooxygenation reactions or superoxide dismutation (Table 1). Intriguingly, except plastocyanin, located in the thylakoid lumen [26] and required for electron transport processes, most cuproenzymes identified so far are localized to the bacterial membrane, the periplasmic space or the cell surface. The almost complete absence of cytosolic cuproenzymes in bacteria might be one strategy to prevent Cu toxicity simply by compartmentalization. This is also exemplified by the fact that almost all studied cuproenzymes appear to be metalated outside of the cytosol [25,27,28]. Even for the periplasmic multi copper oxidase CueO, which folds partially inside of the cytosol and is secreted by the Tat protein transport pathway [29], Cu insertion likely occurs in the periplasm [30]. However, bacterial cells contain a cytosolic Cu pool of often poorly defined nature. For example, detailed studies on the Cu delivery pathway for the *cbb*_3_-type cytochrome *c* oxidase (*cbb*_3_-Cox), have demonstrated that assembly and activity of *cbb*_3_-Cox is strictly dependent on the P_1B_-type Cu-exporting ATPase CcoI [31,32]. Thus, even though Cu is inserted into the catalytic heme *b*-Cu_B_ center of *cbb*_3_-Cox from the periplasmic side of the membrane [33], Cu delivery still depends on an obligatory cytosolic Cu pool. One possible explanation for this puzzling observation is that Cu is preferentially inserted into cuproenzymes as chaperone-bound Cu(I) and routing it through the cytoplasm, ensures control of its reduced state as Cu(I).

Cu insertion into cuproenzymes is generally facilitated by dedicated auxiliary Cu-chaperones, such as NosL for nitrous oxide reductase [44,45] or the ScoI- and PCu_A_C-like chaperones for Cox [46,47]. Intriguingly, in some cases, these Cu-chaperones can be bypassed by increasing the Cu-concentration in the medium, suggesting that they are particularly important at low Cu availability. In bacteria, Cu-chaperones can also act on different targets. The ScoI- and PCu_A_C-like chaperones were initially linked to the assembly of the periplasm-exposed binuclear Cu_A_ site in subunit II of *aa*_3_-Cox [48,49,50,51]. However, it is now evident that both proteins are also involved in the formation of the deeply membrane-buried Cu_B_-site of subunit I of *cbb*_3_-Cox [46,47,52,53].

Cu binding sites within proteins are usually composed of cysteine, methionine, histidine and occasionally tryptophane residues [54]. The composition and geometry of the Cu binding motif is important for its reduction potential. While redox cycling is required for the catalytic activity of cuproenzymes, cuproproteins involved in Cu trafficking need to avoid Cu(I)/Cu(II) redox cycling and therefore their Cu binding motifs require a more negative reduction potential that stabilizes the Cu(I) state [54] (Figure 2). Copper trafficking proteins coordinate Cu(I) via a CxC or CxxC motif with a linear geometry as in Cu(I)-transporting P_1B_-type ATPases or in CopZ-like Cu chaperones [54], or with a trigonal planar geometry as in Sco1-like chaperones. Here, two cysteine residues and a distal histidine residue provide the binding site for Cu(I). In cuproenzymes, such as Cu,Zn superoxide dismutase (SOD) or cytochrome oxidase, Cu is coordinated by histidine, cysteine and methionine residues (Figure 2). Cu(II) is preferentially liganded via histidine nitrogen donors in a square planar arrangement [55,56,57]. Methionine as ligand is less pH sensitive and more hydrophobic and provides often an additional weaker ligand for Cu [58].

In this review, we summarize the current knowledge on how Cu is transported across the outer and inner membranes in bacteria, how these transport processes are linked to cuproenzyme assembly pathways and how a sophisticated protein network guarantees cellular Cu homeostasis providing sufficient Cu supply for cuproenzyme biosynthesis while simultaneously preventing Cu toxicity. The players involved in this network will be described following the route that Cu takes all the way from the environment to their respective cuproenzymes.

## 2. Copper Import across the Outer and Inner Membranes in Bacteria

Since Cu is an essential micronutrient, its passage across biological membranes is crucial for all organisms. In eukaryotes, Cu import is mainly mediated by members of the Ctr (copper transport) family of transporters [59]. Eukaryotic cells contain between one and six Ctr family members (also known as SLC31 family), which display some subcellular or organ specificity [59]. Especially Ctr1 is well characterized in terms of its physiology, structure and function [60,61,62,63]. Cytosolic Cu is further distributed into eukaryotic organelles by P_1B_-type ATPases (also known as CPx-ATPases) [64]. In contrast to the Ctr family members, which so far have not been identified in bacterial genomes [62], P_1B_-type ATPases are universally conserved and widely distributed in different bacterial species [1]. As in eukaryotes, bacterial P_1B_-type ATPases export Cu out of the cytosol and are apparently not involved in Cu import. Although some P_1B_-type ATPases, such as CtaA of the cyanobacterium *Synechocystis* PCC 6803 and CopB of *Enterococcus hirae*, were initially suggested to import Cu into the cytosol, further analyses indicated that both proteins are involved in Cu export out of the cytosol [65,66,67,68]. In cyanobacteria, a second P_1B_-type ATPase, PacS, distributes Cu further into the thylakoid lumen for plastocyanin and *caa*_3_-Cox assembly [69,70,71].

Bacterial Cu importing systems in general are more diverse than in eukaryotes and some of the well-studied mechanisms of Cu uptake are limited to one group of bacteria, or even to one species. This section will mainly focus on Cu import mechanisms in Gram-negative bacteria, where most of the well-studied Cu-uptake pathways have been described.

### 2.1. Cu-Uptake across the Outer Membrane of Gram-Negative Bacteria and Mycobacteria

The passage of ions and small molecules across the outer membranes generally occurs either via various porins or through energy-dependent mechanisms. Transcriptional profiling and comparative differential cuproproteomics using *Pseudomonas aeruginosa* [72], *Acidithiobacillus ferrooxidans* [73] and *Rhodobacter capsulatus* [74] revealed that the expression or steady-state levels of several outer membrane proteins changed when cells were exposed to Cu stress. For most of these proteins, a role in Cu uptake still remains to be defined, but some studies have identified specific outer membrane proteins with more defined role(s) in Cu uptake (Figure 3).

*E. coli* mutants lacking the outer membrane porin OmpF were shown to be Cu resistant, suggesting a role of this porin in Cu(II) uptake [75]. However, various isogenic *ompF* and *ompC* mutants of other *E. coli* strains did not show significant changes in Cu resistance, suggesting the presence of additional Cu import systems [76]. The involvement of porins in Cu uptake has also been shown in mycobacteria, which contain a complex cell envelope that consists of the cytoplasmic membrane, the peptidoglycan–arabinogalactan complex and an outer membrane that is covalently attached to arabinogalactan [77,78]. In *Mycobacterium smegmatis*, mutants lacking the porins MspA or MspC show severe growth defects on Cu-limited growth media, and increased Cu tolerance when grown at high Cu concentrations [79]. Heterologous production of *M. smegmatis* MspA suppresses growth of *M. tuberculosis* at high Cu concentration, further indicating that MspA is involved in Cu uptake across their outer membrane [79,80,81]. The X-ray diffraction analysis of *M. smegmatis* MspA showed an interconnected octamer with eightfold symmetry that resembles a goblet with a single central channel [80]. A recent study, searching for new strategies to boost the antimicrobial activity of Cu, identified 8-hydroxyquinoline (8HQ) as a potent Cu-dependent bactericidal of *M. tuberculosis*. However, the antimicrobial activity of 8HQ–Cu was also observed in a Δ*mspA* mutant strain of *M. smegmatis*, suggesting that MspA is not essential for the uptake of the 8HQ–Cu complex [82].

NosA is an outer membrane protein that is involved in the biogenesis of the periplasmic, Cu-containing enzyme nitrous oxide reductase (NosZ). NosZ contains two Cu centers (Cu_A_ and Cu_Z_), which are required for N_2_O reduction [83,84]. NosA was first identified in *Pseudomonas stutzeri* in a series of mutants producing inactive NosZ variants that lacked Cu in the catalytic center [85]. NosA shows some homology to known *E. coli* TonB-dependent outer membrane transport proteins, including BtuB, TutA, FepA and FhuA [86]. TonB-dependent transport systems are unique in bacteria and use the electrochemical potential of the cytoplasmic membrane for transporting nutrients across the outer membrane [87,88,89,90]. NosA is likely required for NosZ assembly at low Cu concentrations and might also function as a general Cu import system, rather than dedicated specifically to NosZ assembly [91,92]. Nonetheless, NosA-dependent Cu uptake is still considered as one of the main routes for Cu delivery to NosZ in NosA-containing organisms, but alternative pathways for NosZ metalation are suggested to exist in organisms lacking NosA [18].

**Figure 3 membranes-10-00242-f003:**
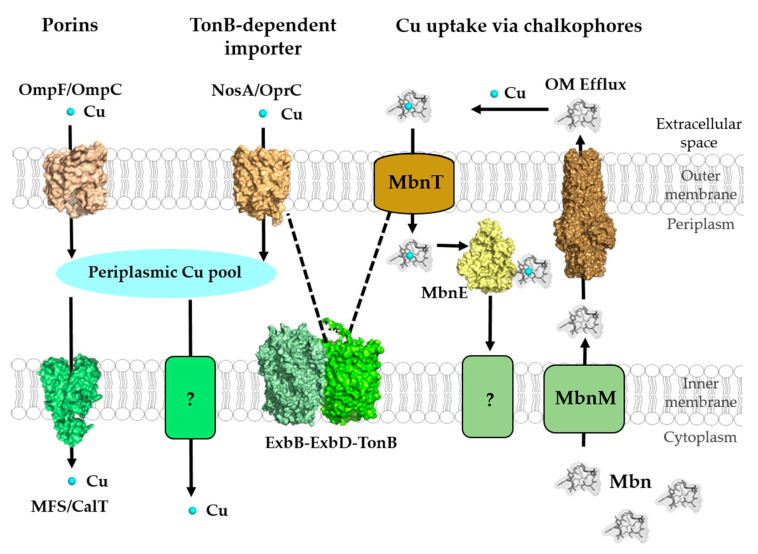
Cu-transport across the outer and inner membranes of bacteria. Bacteria use three established systems for Cu transport across the outer membranes: porins, TonB-dependent transport systems, such as NosA/OprC, and uptake via chalkophores, such as Methanobactin (Mbn), which represent Cu-specific metallophores. Periplasmic Cu is transported into the cytoplasm via the CcoA-like transporter (CalT), which belongs to the major facilitator superfamily. Additional Cu import systems (e.g., YcnJ-like proteins [93]) likely exist in bacteria, but have not been characterized [23]. Mbn is synthesized in the cytoplasm and transported via MbnM into the periplasm and by a so far unknown transporter into the extracellular space. Cu-loaded Mbn is imported back into the periplasm via the TonB-dependent transporter MbnT and bound by the periplasmic protein MbnE. How Cu-Mbn is transported back into the cytosol across the cytoplasmic membrane and how Cu is released from Mbn in the cytosol are unknown. Dashed lines indicate the interaction of the TonB-dependent transporter NosA/OprC and MbnT with the inner membrane ExbB–ExbD–TonB complex. Structures were retrieved from the protein database (PDB) with the following IDs: 2ZFG (OmpF for Porin), 3WDO (YajR for CalT/CcoA), 3j09 (CtaA), 4ZA1 (NosA), 5ZFP (ExbB/ExbD hexameric complex), 5ICQ (MbnE), 1EK9 (OM efflux), 2XJH (Mbn), and depicted using Pymol.

OprC is a TonB-dependent transporter in the outer membrane of *Pseudomonas aeruginosa*. OprC is homologous to NosA of *P. stutzeri* (65% amino acid sequence identity) and binds Cu(II) with micromolar affinities [94]. The expression of OprC was shown to be repressed by high exogenous Cu(II) concentrations and enhanced under anaerobic conditions in the presence of nitrate [94,95]. Although, OprC expression is repressed under Cu stress, it is induced by low Cu(II) concentrations and is directly regulated via the Cu-responsive transcriptional regulator CueR in *P. aeruginosa* [96]. Moreover, OprC is an important determinant for bacterial competition and virulence [96]. Very recently, the crystal structures of OprC wild-type and mutant proteins were resolved in the presence and absence of silver and Cu [97]. The structures. as well as the inductively coupled plasma mass spectrometry (ICP-MS) and electron paramagnetic resonance (EPR) data, suggested that Cu(I) binds to a CxxxM-HxM motif. It was furthermore suggested that OprC also binds Cu(II) and is able to reduce it to Cu(I) via thiol groups, although this awaits further validation [97].

Unlike NosA and OprC, the small outer membrane protein ComC (also known as YcfR or BhsA) acts by lowering the permeability barrier of the outer membrane to Cu. Most Gram-negative bacteria encode ComC-like proteins with 50% to 90% sequence homology. In the absence of ComC, *E. coli* shows reduced Cu import into the cytoplasm [98]. The transcription of *comC* is induced via the TetR-like transcriptional regulator ComR in response to Cu availability [67,98]. Initially, ComC was described as a general stress response protein [99,100,101,102] and it remains unclear how ComC is simultaneously involved in different cellular stress response pathways, including Cu stress.

### 2.2. Cu Transit through the Periplasmic Space in Gram-Negative Bacteria

Once Cu has crossed the outer membranes into the periplasm, it is likely scavenged by periplasmic Cu chaperones, Cu-storage proteins and chemical chelators, such as glutathione [18,103,104]. Although several periplasmic Cu chaperones have been described in multiple bacteria, they are mainly associated with Cu-detoxification and cuproenzyme biogenesis pathways [47,52,105]. Hence, it is currently not clear, whether they also participate in Cu uptake, and therefore these proteins are described later. Dedicated Cu-storage proteins (Csps) were first described in the Gram-negative methanotrophic bacteria, but are widely distributed in bacteria [103]. Csp1-type proteins contain a Tat signal sequence and are likely exported into the periplasm for Cu binding [103,106,107]. As cytosolic Csp3-type proteins are much more abundant than the secreted Csp1-type proteins [103], they are further discussed in Section 4.4.

### 2.3. Cu Uptake across the Inner Membrane

#### The CcoA-Like Cu-Transporter (CalT) Family

Major-facilitator-superfamily (MFS)-type transporters belong to a large and ubiquitous superfamily of transporters. MFS proteins selectively transport a wide range of substrates by using the proton gradient as driving force [108]. CcoA belongs to the newly discovered CalT (*CcoA-like transporter*) family of MFS-type transporters (Figure 1), and is the prototype of a bacterial inner membrane Cu importer [23,33,109,110]. CcoA is the first Cu uptake transporter identified in bacteria and the first MFS-type transporter known to transport Cu. It was first discovered in *R. capsulatus* by genetic screening for *cbb*_3_-Cox defective mutants that could be rescued by exogenous Cu supplementation. Furthermore, heterologous expression of *ccoA* restores the respiratory defect and Cu import in a *Schizosaccharomyces pombe* double mutant that lacked the Cu-importer Ctr4 and Ctr5 [111]. Overall, these studies demonstrated that CcoA is a Cu importer required for the biogenesis of *R. capsulatus cbb*_3_-Cox. Interestingly, bypass suppressors of *ccoA* deletion mutants that suppress the *cbb*_3_-Cox defect were frequently observed in *R. capsulatus* [23,109]. These bypass suppressors were Cu sensitive and had higher cellular Cu content compared to wild-type and *ccoA* mutant strains. Whole genome sequencing revealed that these suppressor mutants contained single base pair insertion/deletion in *copA* [109], encoding the well-known P_1B_-type Cu exporter, CopA, involved in Cu detoxification [25,112]. These observations indicated the presence of a functional interplay between the Cu importer, CcoA, and the Cu exporter, CopA, in controlling intracellular Cu homeostasis to avoid toxicity and ensure delivery of Cu for *cbb*_3_-Cox assembly in *R. capsulatus*. Intriguingly, studies in *R. sphaeroides* indicated that CcoA is dedicated solely to the biogenesis of *cbb*_3_-Cox, but not required for the biogenesis of *aa*_3_-Cox. Thus, two distinct Cu delivery pathways operate for Cu insertion into two similar cuproenzymes [110].

To uncover the molecular mechanisms of Cu binding to CcoA, sequence alignments of *R. capsulatus* CcoA with other proteobacterial homologues identified two well conserved motifs, M_233_xxxM_237_ and H_261_xxxM_265_ (*R. capsulatus* numbering) [113], predicted within transmembrane domains (TMDs) 7 and 8. Mutations within both putative metal-binding sites block Cu import and *cbb*_3_-Cox assembly [113] (Figure 4).

A comparative genomic analysis of orthologues in α-proteobacterial species showed that CcoA-like homologues are widespread among these organisms, and frequently co-occur with Cox enzymes [110,114]. CalT family members also include the RfnT proteins [114], earlier suggested to transport riboflavin [115,116], but now shown to transport Cu [114]. However, the RfnT-like proteins are unable to restore the *cbb*_3_-Cox defects in *R. capsulatus ccoA* mutants. The lack of functional complementation between CcoA and RfnT-like proteins suggests that even though Cu may be imported via those proteins, its further use is determined by the functional interaction between the Cu importer and the down-stream Cu-binding proteins. Future studies are required to characterize the Cu uptake mechanism and specificity of the CalT family proteins in proteobacteria.

### 2.4. Cu Uptake via Chalkophores: Methanobactins and Yersiniabactin

Under metal-limiting conditions, most bacteria synthesize metallophores, which are organic compounds with high-metal affinity [88,89,117]. Upon their synthesis, the metallophores are exported to the extracellular environment, where they bind their specific metal. The metal–metallophore complexes are subsequently imported back into the intracellular space for metal release. Chalkophores (chalko: Greek word for Cu) are Cu-binding metallophores [117], that are functionally comparable to the well-known siderophores and are synthesized in response to low iron availability [88]. Well-characterized chalkophores are methanobactins and yersiniabactin, which are described in this section (Figure 3). Other organic molecules, such as coproporphyrin III [118], schizokinen [119] or staphylopine [120], might also serve as potential chalkophores in Gram-negative or Gram-positive bacteria [117].

#### 2.4.1. Methanobactin (Mbn)

Mbns were first discovered in methanotrophic bacteria that carry out aerobic hydroxylation of methane to methanol via either the soluble methane monooxygenase (sMMO, an iron-containing enzyme) or the particulate methane monooxygenase (pMMO, a membrane associated Cu-containing enzyme) [35]. Cu-resistant mutants of *M. trichosporium* OB3b showed a drastic increase in non-perceptible Cu suggesting that this species excretes a compound (Mbn) for Cu uptake [121]. The crystal structure and NMR analysis of Mbn showed that this molecule binds a single Cu ion, coordinated by two thioamide or enethiol groups and two neighboring oxazolone rings of chromophoric residues [122,123,124]. Similar structures of additional Mbn were also solved and showed differences in a few residues [125,126,127]. Apo-Mbn binds both Cu(II) or Cu(I), although the final complex contains primarily Cu(I), indicating that Mbns might be able to reduce Cu (II) [127,128]. Mbns have very high affinity to Cu(I) (approximately 10^−20^ M), in line with their role in extracellular Cu acquisition [127].

MbnA is a short peptide (22 to 40 amino acids) that serves as precursor of Mbn in *M. trichosporium* OB3b [129], and is encoded by *mbnA* present in all *mbn* gene clusters [130]. Together with *mbnB* and *mbnC, mbnA* constitutes the core of the *mbn* gene clusters. The MbnBC complex is involved in Mbn biosynthesis, but additional genes can be part of *mbn* gene clusters and are involved in biosynthesis, regulation and Mbn transport [131,132]. Of these, *mbnM* encodes a putative inner membrane efflux pump, which might be involved in the secretion of Mbn after its synthesis [133,134] (Figure 3). Apo-Mbns are transported across the outer membrane by some unknown mechanism, before they are loaded with Cu and taken back up into the cell for Cu release [132]. Uptake of the Cu-Mbn complex is inhibited when the proton gradient is inhibited, indicating energy-dependence for transport into the cytoplasm [135]. The *mbnT* gene encodes a TonB-dependent outer membrane transporter, and although an *M. trichosporium* OB3b *mbnT* mutant is capable to synthesize Mbn, it is deficient for its internalization [136]. After import into the periplasm, Cu-loaded Mbn is bound to periplasmic binding proteins, including MbnE, which is related to the oligopeptide transporter OppA/AppA [117]. However, the further import of Cu-Mbn into the cytoplasm and the mechanisms of Cu release from the Cu-Mbn complex are not yet completely understood (Figure 3).

#### 2.4.2. Yersiniabactin (Ybt)

The Ybt system is a well-known siderophore-dependent iron transport system discovered in some pathogens such as *Yersinia* sp. and enterobacteria. In *Yersinia*, it is encoded by a multi-operon high pathogenicity island [137,138]. In an attempt to screen for other metals that are bound by Ybt, an ICP-MS approach revealed that Ybt can also bind Cu(II) and that the formed Ybt-Cu(II) complex is stable even in the presence of Fe(III) [139,140]. The Ybt–Cu(II) complex also exhibits superoxide dismutase-like activity that potentially protects the pathogen against the respiratory burst-derived superoxide in host phagocytes [141]. The mechanisms of Ybt–Cu transport and Cu release are largely unknown. A TonB-dependent outer membrane siderophore uptake protein, FyuA (for ferric-Ybt uptake) is located in the high pathogenicity island and *E. coli* Ybt-deficient mutants expressing FyuA accumulate exogenously supplied Ybt–Cu(II) in the cytosol [142,143,144]. This supports a role of FyuA in Ybt-metal complex uptake. The subsequent metal release, might require the inner membrane-bound ATP-binding cassette transporter, YbtPQ [142].

## 3. Reduction of Cu(II) to Cu(I) Is a Prerequisite for Cytoplasmic Copper Storage and for Re-Routing It to the Periplasm

The known bacterial cytosolic Cu-chaperones and Cu-exporting P_1B_-type ATPases preferentially bind and export Cu(I). The mechanisms for binding and exporting Cu likely evolved under the conditions of the primordial anaerobic atmosphere, in which Cu was mainly present in its reduced Cu(I) form. Presumably, these export systems were initially required exclusively to prevent Cu toxicity and were then later on adopted for cuproenzyme biogenesis under aerobic conditions [145], making a cytosolic Cu(II) reduction step mandatory in bacteria. Reduction of Cu(II) is generally favored by the reducing environment of the cytosol, which contains chemical reductants, including glutathione, quinols, cysteine or ascorbic acid. Furthermore, non-specific metal reductases can reduce Cu(II) to Cu(I) once Cu enters the cytosol [146,147,148]. However, this process may be rate-limiting.

This process of Cu(II) reduction is different in eukaryotes, which reduce Cu(II) on the cell surface by extracellular cupric reductases before copper import [149]. The responsible enzymes are primarily heme-dependent iron reductases (Figure 5) that can also reduce Cu(II) and use NADPH or ascorbic acid as electron donor [150,151,152,153,154]. Intriguingly, it was recently reported that a histone H3-H4 tetramer has NADH-dependent cupric reductase activity [155]. Histone proteins are ubiquitous in eukaryotic cells and required for genome organization. The physiological significance of this activity is not known, although it may provide Cu(I) needed for some cuproenzymes, e.g., superoxide dismutase [155]. It was speculated that histone proteins might have evolved to adapt to increasing O_2_ concentrations in the atmosphere [156].

Despite the need for Cu(II) reduction in the bacterial cytosol, Cu-specific reductases have not been identified in bacteria until recently [157]. It was generally assumed that reducing agents, such as glutathione and ascorbic acid, are sufficient for providing enough Cu(I) for further transfer to Cu-chaperones [146,148]. In particular, the abundant and ubiquitous glutathione was thought to be an important player in Cu reduction [158]. However, glutathione-dependent Cu(II) reduction has some disadvantages, because the glutathione-Cu(I) complex can spontaneously oxidize back to Cu(II) and convert O_2_ to superoxides, increasing the risk of oxidative damages [146]. The presumed absence of specific cupric reductases in bacteria is also inconsistent with the fact that all other individual steps of cellular Cu homeostasis are usually tightly regulated and controlled for decreasing the risk of oxidative damage.

The membrane-bound protein CcoG from the proteobacterium *R. capsulatus* was recently characterized as the founding member of a new and widespread class of cupric reductases in bacteria [157]. CcoG was initially identified as a putative assembly factor of *cbb*_3_-Cox**, because it is encoded in the highly conserved *ccoGHIS* (also called *fixGHIS*) gene cluster, located immediately downstream of the structural genes of *cbb*_3_-Cox. Early studies have shown that CcoH, CcoI and CcoS are essential for the production of active *cbb*_3_-Cox [31,32,159,160,161], but no function was assigned to CcoG. Further characterization of CcoG revealed the presence of two tetranuclear iron–sulfur clusters and two Cu binding sites (Figure 5) and in vitro data demonstrated that CcoG is able to reduce Cu(II) but not Fe(III) in the presence of an artificial electron donor. In the absence of CcoG, *cbb*_3_-Cox activity and the Cu_B_-containing catalytic subunit CcoN are severely reduced. Furthermore, the absence of CcoG increases Cu-sensitivity, while its over-production promotes Cu tolerance [157]. These findings suggest that CcoG provides Cu(I) not only for *cbb*_3_-Cox assembly but also for Cu-chaperones, such as CopZ, and to P_1B_-type ATPases for Cu(I) export to reduce Cu toxicity. Intriguingly, CcoG and its homologues are also found in bacterial species lacking *cbb*_3_-Cox, such as YccM of *E. coli*. Homologues like YccM are smaller than CcoG as they lack the immunoglobulin (Ig)-like periplasmic domain at the C-terminus of CcoG. The function of the Ig-like domain in CcoG is currently unknown and requires further analyses [157].

Except CcoG, there are only few reports on Cu-reducing activities encountered in bacteria (Table 2). For example, NADH dehydrogenase-2 in *E. coli* was shown to reduce Cu in the presence of NADH or FAD [162]. However, this activity was later attributed to unspecific reduction via quinols, similar to a Cu(II) reducing activity described in *Lactococcus lactis* [163]. A Cu-reducing activity was also detected in cell-free extracts of Cu-resistant *Pseudomonas* species, but not further characterized [164]. Finally, a Cu-reducing activity was observed for the Cu-chaperone CupA in vitro, but the main function of this protein appears to transfer Cu(I) to the P_1B_-type ATPase CopA [165,166]. Thus, to the best of our knowledge, CcoG is the first specific bacterial cupric reductase.

In archaea, so far the only known Cu(II) reduction activity is exhibited by the Cu-chaperone CopZ of *Archeoglobus fulgidus*. CopZ-like Cu-chaperones are conserved among all domains of life (see Section 4.1), but the *A. fulgidus* CopZ is structurally unique, because it is fused to an iron–sulfur-cluster binding domain that mediates Cu(II) reduction in the cytosol [167].

The available studies suggest that the mechanism of Cu(II) reduction reactions used by eukaryotic, archaeal and bacterial Cu(II) reductases differs greatly (Figure 5).

## 4. The Cytosolic Cu Pool: Chaperones, Storage Proteins and Chemical Chelators

The virtual absence of free Cu in the bacterial cytosol is achieved by Cu-chaperones, Cu-storage proteins and low-molecular weight thiol-containing molecules, which scavenge Cu and transfer it subsequently to Cu-efflux systems or to Cu-containing proteins [103,158,168,169]. Affinity gradients are important determinants for vectorial Cu transfer between these donor/acceptor components (Table 3) [169].

### 4.1. The CopZ-Like Chaperones

CopZ-like chaperones are universally conserved and ubiquitous small proteins (~7–8 kDa), composed of approximately 70 amino acids. CopZ from the Gram-positive bacterium *E. hirae* was the first characterized bacterial Cu-chaperone and shown to be required for copper resistance and copper export [170,171]. In bacteria, CopZ genes are often, but not always [172], encoded in one operon together with the Cu-exporting P_1B_-type ATPase CopA and Cu-sensing transcriptional regulators [173,174]. In *E. coli*, a CopZ chaperone is synthesized upon ribosomal frame-shifting during translation of the *copA* mRNA [175]. The high efficiency of frame-shifting is achieved by the combined stimulatory action of a “slippery” sequence, an mRNA pseudoknot, and the CopA nascent chain [176]. Binding affinities for Cu(I) have been determined for CopZ and its eukaryotic homologues Atox1 and are in the range of 10^−22^ M [169,177] (Table 3), rationalizing why free Cu is basically undetectable in the bacterial cytosol. Apo-CopZ is a monomer and contains a typical ferredoxin-like (βαββαβ) fold, in which four β-strands and two α-helices form a double sandwich (Figure 6). The same βαββαβ-fold is also found in the N-terminal metal-binding region of P_1B_-type ATPases, allowing Cu(I) transfer from CopZ to the Cu-exporting ATPases [178,179,180] (see Section 5.1). The MxCxxC Cu(I)-binding motif of CopZ is located in a loop connecting the first β-strand with the first α-helix [171]. Cu(I)-binding to CopZ has been shown to induce dimerization [181,182], in which Cu(I) is present as a tetra-nuclear Cu complex in a Cu_4_(CopZ)_2_ stoichiometry [182]. The physiological role of CopZ dimers is not entirely clear and dimerization might only occur at high Cu concentrations [172,183]. Furthermore, dimerization is inhibited by glutathione, and high cellular glutathione concentrations possibly prevent CopZ dimerization in vivo [145,184].

The cytoplasm of *P. aeruginosa* contains a Cu-sensing transcriptional regulator, CueR (see Section 5.4), and two homologous metal chaperones, CopZ1 and CopZ2, sharing 37% sequence identity [185]. Analysis of 896 homologous sequences showed that CopZ-like chaperons can be divided into two subgroups: the CopZ1-like subgroup (544 sequences with a conserved CxGC motif, including eukaryotic CopZ-like chaperones) and CopZ2-like proteins (352 sequences with an invariant histidine residue in the conserved MxCxHC motif, which are only present in prokaryotes). Cu(I) exchange between both chaperones is rather slow and kinetically restricted. This finding indicates the presence of two separate Cu pools in *P. aeruginosa*, bound to either CopZ1 or CopZ2. Although both chaperones can transfer Cu(I) to CueR in vitro, experiments in vivo indicate that only CopZ1 metallates CueR, consistent with the relative K_D_ values of the three proteins. The more abundant CopZ2 was proposed to serve as a Cu storage pool, providing a fast-homeostatic response to high Cu levels, while excess Cu(I) is transferred to CopA [185].

The proteobacterium *R. capsulatus* contains only one *copZ*, which is required for both Cu resistance and full assembly of *cbb*_3_-type Cox [172]. This dual role of CopZ is supported by the observation that it can be detected in a complex with either CcoI or CopA in *R. capsulatus* membranes [172]. Thus, CopZ supplies Cu not only to CopA-dependent detoxification but also to cuproprotein biogenesis via the alternative Cu(I)-exporting ATPase CcoI, dedicated to *cbb*_3_-Cox assembly [172].

### 4.2. CupA

The membrane-anchored Cu-chaperone CupA is essential for Cu resistance in *Streptococcus pneumoniae* and found in only a few other streptococcal species [165]. Cu sensitivity of a Δ*cupA* strain is comparable to that of a Δ*copA* strain [166,186], but a Δ*copA* strain is much less virulent than a Δ*cupA* strain in a mouse model, suggesting that Cu export via CopA is not strictly dependent on CupA in this system [187].

The structure of CupA shows a cupredoxin-like eight-stranded β-barrel containing a binuclear Cu cluster, and a single cysteine residue as bridging ligand between the two copper sites, S1 and S2 [186] (Figure 6). By using X-ray absorption spectroscopy and NMR, the bis-thiolate S1 site could be determined as the high affinity Cu(I) site, while the methionine-rich site constitutes the low-affinity S2 site [186]. The ability of CupA to bind Cu(II) at the S1 site, enables it to trap Cu at the cytoplasmic membrane after Cu(II) has entered the cell. In vitro, CupA extracts Cu(II) from the transcriptional repressor CopY and the apo-CopY represses the *cop*-operon in this species [165]. CupA can also reduce Cu(II) to Cu(I) via the cysteine residues of the S1 site in vitro [165]. Reduced Cu(I) can then be transferred to the S2 site and handed over for export to CopA. CopA in *S. pneumoniae* contains a cupredoxin-like N-terminal metal-binding site, instead of the typical ferredoxin-like domain that is found in CopZ and the N-terminal metal-binding sites of other CopA-like proteins [186]. Interestingly, the function of CupA cannot be replaced by a chimeric construct consisting of *B. subtilis* CopZ tethered to the N-terminal TM-helix of CupA. These findings underscore the importance of protein–protein interactions between the Cu-chaperones and their cognate P_1B_-type ATPases for Cu export. The occurrence of either ferredoxin–ferredoxin or cupredoxin–cupredoxin domains on protein pairs that specifically interact to ensure cytoplasmic Cu export represent an exquisite example of convergent evolution [188].

**Table 3 membranes-10-00242-t003:** Cu affinities of bacterial Cu-chaperones and metal-binding sites (MBS) of P_1B_-type ATPases. N-MBS refers to the N-terminal MBS and TM-MBS to the membrane-integral MBS of P_1B_-type ATPases. Bacterial P_1B_-type ATPases often have two N-MBS.

Protein/Organism	Affinity for Cu(I) (K_d,_ M)	Reference
CopZ (*B. subtilis*)	6 × 10^−22^	[189]
N-MBS1 (*E. coli*)	7 × 10^−19^	[189]
N-MBS2 (*E. coli*)	3 × 10^−18^	[189]
TM-MBS1 (*A. fulgidus*)	1 × 10^−15^	[190]
TM-MBS2 (*A. fulgidus*)	1 × 10^−15^	[190]
CusF (*E. coli*)	5 × 10^−11^	[191]
PccA (*R. capsulatus*)	8 × 10^−16^	[47]
SenC (*R. capsulatus*)	3 × 10^−15^	[47]

### 4.3. Metallothioneins

Metallothioneins (MTs) constitute a superfamily of cysteine-rich small proteins that bind multiple metal ions, including Cu(I), with high affinity [169]. MTs have been characterized by the presence of single or multiple metal-thiolate clusters in their structures [192,193]. Their roles include chelation, intracellular distribution, storage and detoxification of metals and defense against oxidative stress [194]. Eukaryotic MTs usually contain 20 cysteine residues that can bind up to seven metal ions per MT molecule. While eukaryotic MTs have been extensively characterized, bacterial counterparts are rare, but present in both Gram-positive and Gram-negative bacteria [195,196,197]. Bacterial MTs have been first identified in the marine cyanobacterium *Synechococcus sp*. [198]. SmtA from *Synechococcus* PCC 7942 was shown to be involved in Zn(II) and Cd(II) sequestration in vivo. Its structure shows a zinc finger-fold that is not common for eukaryotic MTs. Cu(I) binding of SmtA was observed in vitro, but it is unclear whether SmtA is involved in Cu detoxification in vivo [199]. Homologues of SmtA have been identified in a limited number of bacterial genomes, ranging from cyanobacteria, pseudomonads, α-proteobacteria, γ-proteobacteria to firmicutes. With the exception of the residues constituting the zinc finger-fold, these homologues show large sequence variations, including their metal-binding sites [199]. In addition to SmtA, screening of a genomic library from *Mycobacterium tuberculosis* identified a small, non-annotated open reading frame (MT0196) that encodes a 4.9-kDa cysteine-rich protein that was later termed MymT. Up-regulation of MymT is required for Cu resistance in *M. tuberculosis* and the protein is predicted to bind up to six Cu(I) ions in a solvent-shielded core [200].

### 4.4. Copper Storage Proteins

Before the discovery of the family of Cu storage proteins (Csps), it was widely assumed that bacteria do not store large amounts of Cu in the cytosol in order to avoid its potentially toxic effects. Csps were first discovered in the methanotrophic bacterium *Methylosinus trichosporium* [201], which produces large amounts of Cu-dependent particulate methane monooxygenase (pMMO) for methane hydroxylation to methanol [202] (see above). Csps are highly upregulated at high Cu concentrations, supporting the idea of Csps being a short-term Cu storage for pMMO biogenesis. Two families of Csps can be distinguished based on their cellular localization: (1) Csp1-type proteins contain a N-terminal Tat signal sequence and are exported into the periplasm or into intracellular compartments of methanotrophic bacteria [203]. Csp1-type proteins are mostly found in methanotrophic bacteria, where they often co-occur with methanobactin, but they are also present in other Gram-negative bacteria, such as *Neisseria gonorhoeae* [103,201,204]; (2) Csp3-type proteins lack an export signal and are localized in the cytosol [204]. The latter family of Csps is found in a wide range of bacteria, archaea and in a few fungal and plant species [103]. Incidentally, the *M. trichosporium* Csp2 is now classified as a Csp1-type Csp, but unlike other Csp proteins it is not up-regulated upon Cu addition [103]. The crystal structure of Csp1 of *M. trichosporium* shows that 13 Cu(I) ions and Csp3 in *B. subtilis* have up to 20 Cu(I) ions per monomer [201,204] (Figure 6). Csps are organized as homo-tetramers with high Cu storage capacity. Unlike the metallothioneins, which fold into their tertiary structure after Cu binding [196], apo-Csps fold into rigid structures and then bind Cu inside the core that is formed by the four-helix bundle [201]. Most of the Cu ions in Csps are coordinated by thiol groups of cysteine residues within the same or different helices. Interestingly, at the hydrophilic end of the four-helix bundle, where uptake and release of Cu(I) take place, a weaker Cu(I) coordination via methionine and histidine residues is found [201,204]. How Csps are loaded with Cu(I) is not completely known. Although methanobactin cannot serve as Cu(I) donor in vitro [204], CopZ was shown to transfer Cu(I) to Csps (Ccsp) in *Streptomyces lividans* [205]. Csp1-type and Csp3-type Csps have similar affinities to Cu(I) (1–2 × 10^−17^ M) [201,204], but their in vitro Cu(I) release rates are highly different. Removal of Cu(I) from *M. trichosporium* Csp1 by the Cu-chelator BCS in vitro takes about 1 h, whereas Cu release of *M. trichosporium* Csp3 or *B. subtilis* Csp3 is drastically slower [201,204]. This is likely explained by different residues at the hydrophilic end of the four-helix bundle, which could influence solvent accessibility providing a kinetic barrier for Cu(I) removal from Csp3. The different rates of Cu release from cytoplasmic and periplasmic Csps could also reflect different affinities of their respective acceptor proteins in the cytosol or periplasm. Moreover, the ability of Csp3 to bind a large amount of Cu(I) in combination with the low removal rate, could be crucial for providing a safe cytosolic Cu storage system. Heterologous expression of Csp3 in *E. coli* enhances Cu resistance and increases cytosolic copper concentration [107]. Therefore, Csps of pathogenic bacteria, such as *S. pneumoniae* and *Salmonella enterica*, could also contribute to their defense mechanisms against Cu intoxication by the host cell [103].

### 4.5. Glutathione as Chemical Cu Chelator

The universally conserved tripeptide glutathione (GSH) is best known for its crucial role in maintaining the cellular redox homeostasis in prokaryotes and eukaryotes [184]. GSH also binds Cu and is connected to Cu homeostasis in bacterial and human cells [158,206]. The formation of Cu(I)-GSH complex was shown in vitro [169,207,208] and one model suggests the formation of a stable (Cu_4_(GSH)_6_) cluster [209]. Binding of Cu(I) to GSH is suggested to serve as a protective mechanism in human cell lines [210]. On the other hand, there is also evidence that Cu(I)–GSH complexes favor the formation of deleterious superoxide in vitro [146]. It is unclear to which extent Cu(I) is ligated to GSH in vivo, because GSH at millimolar concentrations is unable to extract Cu from Cu-chaperones [169]. However, in vivo data suggest that GSH is involved in transferring Cu to Cu-chaperones immediately after it enters the cell [211]. The increased sensitivity of a GSH deficient *S. pneumoniae* strain towards divalent metal ions supports a protective role of GSH against metal stress [212]. In contrast, a GSH-deficient *E. coli* strain shows no increase in Cu-sensitivity unless CopA is also deleted [158]. Studies using a *S. pyogenes* Δ*copA* strain show that mis-metallation in the presence of Cu is reduced when cells grow under GSH supplementation and that GSH competes with CopY for Cu when Cu concentration is high [213]. Thus, GSH that is present at millimolar concentrations inside the cell likely binds Cu primarily when the maximal capacity of Cu chaperones and exporters to bind Cu is consumed. GSH might also buffer the periplasmic Cu pool, because there is evidence for an active transport of GSH into the periplasm by the CydDC transporter [104,214]. However, to what extent periplasmic GSH can substitute for periplasmic Cu chaperones remains unknown.

## 5. Copper Export across the Cytoplasmic Membrane

Exporting Cu from the cytosol into the periplasm or extracellular space is of utmost importance for Cu homeostasis to prevent Cu toxicity and to sustain the biogenesis of membrane-bound and periplasmic cuproenzymes. Accordingly, most studies on bacterial Cu transport focused on understanding Cu export processes.

The dual physiological role of Cu export processes raises questions of how cells rapidly export excess of Cu, while simultaneously providing a continuous flow of Cu dedicated to cuproenzyme biogenesis. One answer to this question lies in the large variety of different types of Cu exporters, which comprise different families: P-type ATPases, ABC transporters, RND proteins (resistance, nodulation and cell division) and possibly membrane-bound multi copper oxidases [1,145,215,216,217,218,219].

### 5.1. P_1B_-Type ATPases

P_1B_-type ATPases belong to the well-characterized and universally conserved superfamily of P-type ATPases. They export transition metals, including Cu, Co, Cd and other thiophilic cations, using ATP-hydrolysis to protect cells against environmental metal stress. Other members of the P-type family are responsible for extruding ions, such as Na^+^/K^+^, H^+^, Ca^2+^, or lipids from the cell [55,64].

P_1B_-type ATPases are classified into six subgroups (P_1B_-1 to P_1B_-7), according to their substrate specificities and structures [64,220] (Table 4). The P_1B_-3-type ATPases were originally classified as Cu(II) exporting ATPases, but later on, they were shown to transport Cu(I) and thus integrated into the Cu(I)-exporting P_1B_-1-type subgroup [220].

Human ATP7A and ATP7B are the best studied examples of the universally conserved P_1B_-type ATPases and are linked to the Cu metabolism disorders Menkes disease and Wilson disease [225,226,227,228]. These transporters have a dual role in Cu homeostasis of eukaryotic cells, because they transport Cu out of the cell and also import Cu into the lumen of organelles [215]. Most bacterial cells often express multiple types of P_1B_-type ATPases of largely non-redundant function. For example, the genome of the Gram-negative facultative phototrophic bacterium *R. capsulatus* contains five putative P_1B_-type ATPases of which CcoI and CopA are Cu(I) exporting P_1B-1_-ATPases, whereas the remaining Rcc01042, Rcc02064 and Rcc02190 proteins are not yet characterized. Rcc02064 and Rcc02190 are likely P_1B-2_-ATPases, while the function of Rcc01042 is unknown due to the lack of typical transmembrane metal-binding motifs. Neither Rcc01042 nor Rcc02190 appear to be involved in Cu homeostasis, because their absence does not influence Cu resistance or *cbb*_3_-Cox assembly [109].

There are two subtypes of Cu(I) transporting P_1B_-type-ATPases in bacteria. CopA1-type ATPases (also known as CopA or CtaA) are responsible for exporting excess Cu(I) [70,109,229]. In contrast, CopA2-type ATPase (also known as CcoI, FixI, CtpA or CopB) transport Cu(I) into the periplasm for subsequent insertion into cuproenzymes [32,230]. In line with their distinct functions, CopA1-type ATPases are characterized by a low Cu(I) affinity, but high turn-over rate, while CopA2-type ATPases have a high affinity and low turn-over rate [70,231].

#### 5.1.1. Structure of P_1B_-Type ATPases

Like all P-type ATPases, the P_1B_-type ATPases are integral membrane proteins that contain three large cytoplasmic domains. The nucleotide-binding domain (N-domain or ATP binding domain) binds ATP and facilitates phosphorylation of an aspartyl residue within the invariant DKTGT motif of the phosphorylation-domain (P-domain). Phosphorylation induces conformational changes within the actuator domain (A-domain), which allows Cu(I) translocation across the membrane. P_1B_-type ATPases usually contain eight TMs (H1 to H8), which are organized in two parts. A core of six TMs (H3 to H8) is present in all P-type ATPases, and preceded by two additional TMs (H1 and H2, also denoted MA and MB) in P_1B_-type ATPases [57,217] (Figure 7).

Two Cu binding sites are present in the transmembrane region of CopA in *A. fulgidus* and *E. coli* [105,180]. The first transmembrane metal-binding site (TM-MBS-1) is composed of two cysteine residues in H4 and one tyrosine residue in H7, while the second TM-MBS-2 is constituted by an asparagine residue in H5 and a methionine and serine residue in H8 [57,70,232]. Both TM-MBS bind Cu(I) with femtomolar affinities and it is assumed that this high affinity prevents the release of Cu(I) back into the cytoplasm prior to translocation [180]. The known 3D structure of the *L. pneumophila* CopA shows a Gly–Gly kink at the cytoplasmic side of H2 [57]. This kink exposes three consecutive and electropositive residues (Met_148_-Glu_205_-Asp_337_) that provide the entrance site for Cu(I) and the platform for the interaction with the Cu chaperone CopZ [233]. Due to the nature of the residues involved, the CopZ–CopA interaction is considered to be weak and transient [234].

The cytosolic A-domain is formed by the first soluble loop that connects H4 and H5. It is primarily composed of β-strands and contains a conserved GE motif, which is required for the dephosphorylation of the conserved Asp residue in the DKTGT motif of the P-domain [235]. The N-domain forms together with the P-domain a large cytoplasmic loop in a mixed αβ structure between H6 and H7. These two domains are linked by a hinge region that contains the ATP binding site in a cleft between them [236].

The N- and C-terminal regions are the most variable domains amongst P_1B_-ATPases and most bacterial variants have one or two heavy metal-binding sites at their N-termini (N-MBS), while eukaryotic proteins, such as the Menkes and Wilson disease associated ATPases ATP7A and ATP7B, have up to six N-MBSs. [215]. In some cases, such as the *A. fulgidus* CopA, an additional Cu binding site is present at the C-terminus and suggested to be involved in regulation [237]. The N-MBSs are characterized by a CXXC motif and they primarily adopt a ferredoxin-like βαββαβ fold that binds Cu(I) [145,238]. The ferredoxin-like fold is very similar to the Atx1/CopZ-like Cu chaperones, and these N-MBSs exchange Cu(I) with cytosolic Cu chaperones. This likely reflects the intricate interaction of the P_1B_-ATPases with their cognate Cu chaperones. Recently, a cupredoxin-like fold was detected in the N-MBS of CopA in *S. pneumoniae*, which binds a binuclear Cu(I) cluster [186]. In this organism, the associated Cu chaperone CupA also shows a cupredoxin fold, further suggesting close interactions between the partner proteins. Mutations close to the predicted Cu(I) entrance side of CopA do not impair Cu export, suggesting that Cu-transfer from CupA to CopA follows a different mechanism in *S. pneumonia* [186].

**Figure 7 membranes-10-00242-f007:**
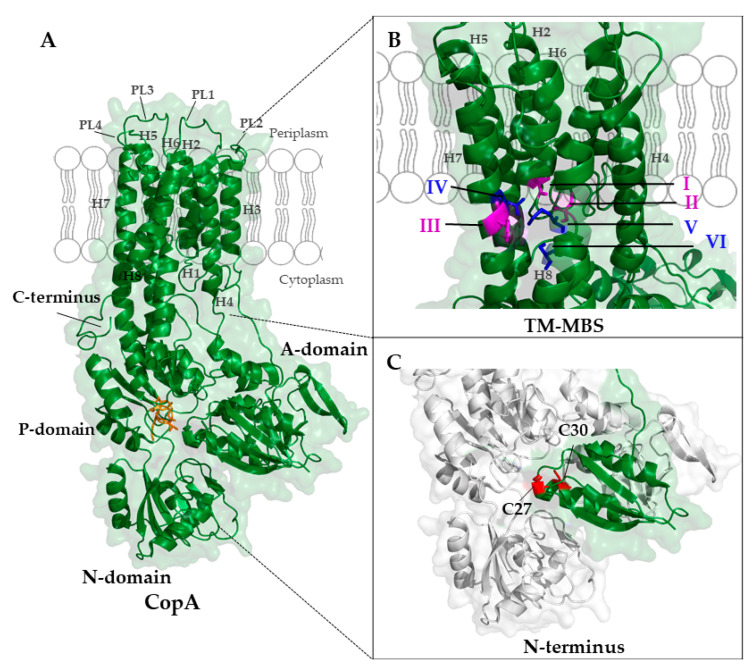
Cryo-electron microscopy structure of the *A. fulgidus* CopA (PDB 3j09) [239]. (**A**) CopA contains eight transmembrane helices (H1-H8), four periplasmic loops (PL1-PL4) and three cytoplasmic domains (actuator domain (A-domain), nucleotide-binding domain (N-domain), and phosphorylation-domain (P-domain)). The invariant DKTGT motif of the P-domain is shown in orange. (**B**) The two transmembrane-metal-binding sites (TM-MBS) of CopA have both trigonal planar coordination geometries and are depicted in magenta (TM-MBS-1; Cys380 (I) and Cys382 (II) in helix 6 and Tyr682 (III) in helix 7)) and blue (TM-MBS-2; Asn 683 (IV) in helix 7, Met711 (V) and Ser715 (VI) in helix 8), respectively. (**C**) The N-terminus of CopA shows a ferredoxin-like βαββαβ-fold (green) with a typical CxxC motif (N-MBS-2); the two cysteines are highlighted in red. Note, that in the *A. fulgidus* CopA structure, the N-MBS-1 is not resolved. The structure of CopA was also determined by X-ray crystallography [57].

#### 5.1.2. Mechanism of Cu Transfer by P_1B_-Type ATPases

Due to the invariant structural motifs, the mechanism of function of the P_1B_-type ATPases probably follow the classical Post-Albers cycle that has been defined for P-type ATPases [240]. During catalysis, the enzyme alternates between two conformations, known as E1 and E2. The E1 conformation is induced by ATP binding to the N-domain and by Cu binding to both the N-MBS and the TM-MBS. ATP-binding leads to the phosphorylation of the conserved Asp residue in the P-domain followed by a conformational change that prevents Cu release back into the cytosol. While transitioning to the E2 conformation, Cu is released into the periplasm. This triggers the dephosphorylation of the DKTGT motif and the enzyme changes its conformation to E2, which can again bind ATP and Cu to switch to the E1 conformation (Figure 8).

Although this mechanism of Cu transfer by P_1B_-type ATPases is generally accepted, the exact function of the N-MBS in Cu transfer is still under debate. In *A. fulgidus* CopA, the N-MBSs are not required for Cu transfer, but they do influence the turn-over rate [241]. In the absence of Cu, the N-MBS was shown to interact with the N-domain [232] and it was proposed that this interaction reduces ATP hydrolysis [242]. Thus, the N-MBS could serve as Cu sensor that represses ATP hydrolysis at low Cu concentrations. At high Cu concentrations, the N-MBS could be loaded with Cu(I) by exchange with the cytosolic Cu chaperone CopZ, and this in turn de-represses ATP hydrolysis [242]. *E. coli* CopA also contains two N-MBS, but both are not essential for function. While the more distal N-MBS was found to behave like a chaperone and could be functionally replaced by a CopZ-like chaperone, the proximal N-MBS was shown to influence the turnover rate in the absence of Cu. The transfer of Cu between the two N-MBS appears to be rather inefficient [189]. N-MBS are also present in CopA2-like ATPases, which bind Cu(I) with high affinity, but have a low turn-over rate. However, the role of the N-MBS in these ATPases has not been studied so far, although interactions of *R. capsulatus* CcoI with CopZ were shown [172], but whether this involves the N-MBS domain remains to be defined. For the eukaryotic ATP7B, it was also shown that Cu(I) binding influences the structure of the N-terminus, which contains six N-MBS [243]. In the absence of five N-MBS the phosphorylation of the P-domain is affected [244]. The first four distal N-MBSs are proposed to be involved in enzyme regulation, while the two proximal N-MBSs are necessary for ATP7B activity [245]. In eukaryotes the function of ATP7B is further regulated by both phosphorylation of the N-MBSs [246], and by Cu-dependent trafficking of ATP7A/ATP7B between the trans-Golgi network and the cytoplasmic membrane [215]. Available data suggest that in the eukaryotic ATP7A/ATP7B, Cu(I) is routed from the cytosolic chaperone Atox1 to the N-MBS, and from there to the TM-MBS [247]. The interaction between N-MBS1 of ATP7A and the human CopZ-like chaperone HAH1 was shown by NMR [248]. This interaction appears to be different for prokaryotic CopA variants, where the chaperone CopZ can deliver Cu(I) to both N-MBSs, but also directly to the TM-MBS via the Cu entrance platform on the cytoplasmic side of H2 [180]. In addition, glutathione and cysteine can also provide Cu for CopA-mediated extrusion in vitro [158,249].

In prokaryotic CopA, both TM-MBSs have to be loaded with Cu(I) before the P-domain can be phosphorylated, and Cu(I) binding to both sites appears to occur independently [250]. Still, TM-MBS-1 is likely occupied first, while TM-MBS-2 is only occupied after ATP binding to the N-domain [190]. Cu-loading of the TM-MBS in *A. fulgidus* CopA appears to be independent of the Cu-occupancy of the N-MBS [180], supporting the conclusion that in prokaryotes Cu is not transferred from the N-MBS to the TM-MBS. From that point on to reach the periplasm, Cu(I) has to be transferred by another, approximately 1 nm, and it was suggested that a dehydrated channel connects the TM-MBS with a periplasmic Cu exit site [251].

Like in the cytosol, free Cu is virtually absent in the bacterial periplasm [25] and CopA does not just release Cu into the periplasm, but it rather transfers it to periplasmic Cu chaperones, such as CusF [105]. CusF is a part of the CusCBA system that translocates Cu across the outer membrane (see below) [252], and it was shown that only apo-CusF interacts with CopA [105]. The key residues involved in this interaction are located predominantly in the negatively charged periplasmic loop P1, which allows docking of the positively charged CusF surface. However, the presence of two TM-MBSs in most CopA homologues also implies that both loading and release occur as a two-step process, and it is not entirely clear how these steps are coordinated. The vectorial, chaperone-driven Cu-export from Cu-loaded CopZ via CopA to Cu-free CusF is further determined by binding affinities and it was proposed that the affinity gradients existing between these components drive Cu distribution [169]. It is, however, important to emphasize that the reported femtomolar affinities for Cu have been determined under non-physiological conditions in vitro through competition experiments and often using only the isolated Cu-binding regions (Table 3). Conformational changes within CopA upon phosphorylation will likely affect Cu(I) affinities and the vectorial Cu transfer will also be influenced by the relative abundance of the interaction partner. Thus, it may be difficult to deduce the directionality of Cu transfer by only comparing the affinities for Cu binding.

CopA also transfers Cu(I) to the periplasmic multicopper oxidase CueO (see below), which oxidizes Cu(I) in the periplasm to the less toxic Cu(II) and thus contributes to Cu resistance [253,254].

In contrast to the CopA1-like ATPases, which export excess Cu(I) for detoxification, CopA2-like ATPases translocate Cu(I) into the periplasm for biogenesis of Cu-dependent proteins. This requires different periplasmic chaperones, and a need for the ScoI-like chaperone SenC and PCuAc-like chaperone PccA for the biogenesis of the *aa*_3_-type Cox and *cbb*_3_-type Cox has been demonstrated in different bacteria [46,47,48,52,53,255]. However, their direct interaction with CopA2-like ATPases for Cu transfer needs to be established.

### 5.2. Cu-Transporting RND Systems: The Cus-System

The Cus resistance system facilitates protection against high levels of Cu(I) in Gram-negative bacteria and has been studied mainly in *E. coli* [256]. It consists of the tripartite CusCBA protein complex, which spans the cytoplasmic membrane, the periplasmic space and the outer membrane. CusA constitutes the substrate binding and transport protein in the cytoplasmic membrane, which is linked to the outer membrane channel CusC via the periplasmic membrane fusion protein CusB. CusA is one of the few members of the heavy-metal exporting resistance-nodulation-cell division protein superfamily (HME-RND) [218]. The RND-like proteins are secondary transporters that act as proton-driven antiporters. In *E. coli*, the CusCBA complex is encoded by the *cusFCBA* operon together with the periplasmic Cu chaperone CusF (see below) [256]. While the deletion of *cusFCBA* has little effect on Cu resistance under aerobic conditions, these mutants are highly sensitive under anaerobic conditions [257]. Expression of the *cusFCBA* operon is regulated by the two-component CusRS system. Absence of the response regulator CusR renders cells highly sensitive to Cu under anaerobic conditions. The change in Cu sensitivity as a function of anaerobiosis versus aerobiosis was rationalized by a dramatic increase in CusC under anaerobic conditions, while the CopA levels remain unchanged and those of the multi-copper oxidase CueO decrease. It appears that under anaerobic conditions, where Cu(I) is the major Cu species, its rapid removal from the periplasm by CusC becomes particularly important [257].

#### Structure and Mechanism of the CusCBA Complex

The crystal structure of CusA revealed a homotrimeric structure in which each monomer consists of 12 TM helices [258] (Figure 9). They are connected by two large periplasmic loops between TM1 and TM2, and TM7 and TM8, respectively. The periplasmic loops can be divided into six sub-domains, a typical feature found in members of the RND superfamily, and also observed in the multidrug efflux pumps AcrB or MexB [259,260]. The subdomains PN1, PN2, PC1 and PC2 form the pore domain, while the subdomains DN and DC form the docking site for CusC. Three adjacent methionine residues (methionine triad) at the bottom of a cleft formed by PC1 and PC2 provide the ligands for binding Cu(I) or Ag(I) [261]. Metal binding leads to a more open cleft between PC1 and PC2, allowing Cu(I) to enter the more distal part of the pore domain. Additional methionine residues are located within the transmembrane part of CusA and just above the methionine triad. All these methionine residues are found inside of the channel, which is formed by the CusA monomer. Accordingly, it was suggested that Cu(I) moves along this methionine ladder stepwise from the cytoplasm to the periplasmic docking domain, before its ultimate transfer to CusC. Mutagenesis studies support the importance of these methionine residues in Cu resistance [256,258,262]. Molecular dynamic simulations indicate that Cu(I) extrusion by CusA is connected to three coordinated movements within the periplasmic cleft, leading to its alternating open and close states [263]. A similar movement has also been observed for AcrB [264].

CusB provides the periplasmic docking protein and consists of four domains: three β-domains and one distal α-helical domain [265]. CusB has been crystallized in multiple conformations, indicating its inherent flexibility. The first β-domain is identical in the available structures and consists of six β-strands that interact with CusA. The second domain shows large differences in these structures and it is suggested that a 20° rotation within the hinge region between domain 2 and 3 is responsible for these conformational changes. The third and fourth domains are largely similar in the different conformations [265,266]. Cu binding to CusB is mediated by an N-terminal methionine triad that bind Cu(I) in a trigonal planar geometry [266]. Co-crystallization of the CusBA complex demonstrated that the trimeric CusA pump is in contact with six CusB proteins, arranged in an upside-down funnel-like structure [267]. The inner surface of the funnel aligns with the periplasmic domains of the CusA trimer providing a continuous conduit with an overrepresentation of negatively charged residues, which could bind Cu(I). The methionine triad of CusB is likely to be the entrance side for Cu(I) delivered by the periplasmic Cu chaperone CusF [268]. Here, the electropositive surface of CusF can interact with the negatively charged N-terminus of CusB [269,270].

The structure of the outer membrane factor (OMF) CusC was also solved and revealed a homotrimeric structure that is typical for the outer membrane component of efflux pumps [271]. Thus, the overall architecture of the CusCBA complex is CusA_3_-CusB_6_-CusC_3_. Each CusC monomer contains four β-strands, which assemble into a 12-stranded outer membrane β-barrel in the trimer. In addition, each protomer contains nine α-helices forming a periplasmic α-barrel in the trimer. Although the function of CusC in the CusABC complex cannot not be replaced by similar OMFs, such as TolC or OprM, CusC does not show any specific Cu(I) binding signatures within the periplasmic or outer membrane barrel. The available methionine residues within CusC appear to be too distant to each other to provide a Cu(I) binding site. However, as with CusB, the internal surface of CusC contains many negatively charged residues which could facilitate Cu(I) extrusion [271].

### 5.3. The Cop/Pco Systems

Many Gram-negative bacteria contain a third line of defense against high Cu concentrations, the *copABCD* or *pcoABCDE* clusters (copper resistance or plasmid-borne copper resistance) [25,272,273]. The *pcoABCDE* cluster is often encoded on transferable plasmids and found in strains experiencing high Cu concentrations in their environment [274,275]. In *E. coli*, the *pco* cluster was initially identified on the conjugative plasmid pRJ1104, which contains the *pcoABCDRSE* cluster [276]. *pcoABCDE* codes for several inner, outer and periplasmic proteins, while *pcoRS* codes for a Cu-responsive two-component regulatory system. As for the Cus-system, PcoRS regulates *pcoABCDE* expression in response to Cu [274]. However, the exact gene cluster organization is not always conserved between the *cop* and *pco* clusters and a *pcoE* homologue is absent in the cop cluster [272]. The Pco system is also present in some Gram-positive bacteria, such as in *B. subtilis*, but here *pcoC* and *pcoD* are genetically fused and the encoded protein is termed YcnJ [93].

PcoA is the major determinant of the *pcoABCDE* induced Cu-resistance. It belongs to the family of periplasmic multi-copper oxidases and can replace CueO in *E. coli* [276]. Like CueO, PcoA contains an N-terminal Tat signal sequence and is likely transported in its folded state from the cytosol into the periplasm [29]. Oxidation of Cu(I) to the less toxic Cu(II) by multi-copper oxidases, e.g., PcoA, seems to be a conserved strategy for preventing Cu toxicity, and might be required in particular under O_2_-limiting conditions.

PcoB is likely located in the outer membrane and predicted to interact with PcoA [274]. PcoB shows some homology to CopB, which may function as an outer membrane Cu transporter [277]. In general, the exact function of PcoB is uncertain, but it might be required for exporting Cu(II) from the periplasm to the extracellular environment [274,278]. For example, deleting *pcoB* in *Acinetobacter baumanii* causes Cu sensitivity [278].

The inner membrane protein PcoD is a predicted copper permease with eight TMs and required for Cu resistance [275,279]. Its over-production renders *P. syringae* hypersensitive to Cu [277] and it was suggested that CopD is involved in Cu uptake across the inner membrane. Similarly, the *Bacillus subtilis* PcoC-PcoD fusion protein YcnJ, is also implicated in Cu import [93]. Indeed, expression of *ycnJ* in *E. coli* allows Cu uptake, supporting this proposed function (Zhang, Khalfaoui-Hassani and Daldal, unpublished). Interestingly, in *M. trichosporium* OB3b, a *copD* homologue is located in close proximity to the gene for the Cu-containing particulate methane monooxygenase pMMO, and it was suggested that CopD is involved in the so-called “Cu switch” to produce this enzyme when Cu is available [280,281]. However, analyses of a *M. trichosporium* OB3b *copCD* mutant indicated that these proteins are apparently not part of the Cu switch and do not seem to be critical for Cu uptake [282]. A putative Cu importing function of PcoD as a strategy against Cu toxicity is not directly intuitive and it was suggested that PcoD imports Cu for the cytoplasmic assembly of the catalytic Cu center of PcoA [274,275]. Folded and assembled PcoA would then be transported into the periplasm via the Tat pathway. However, assembly of Cu-containing periplasmic proteins usually occurs in the periplasm as also shown for the *E. coli* multi-copper oxidase CueO [30]. However, the cytoplasmic or periplasmic origin of Cu in the assembly process, as well as the exact role of PcoD, remains unknown.

PcoC and PcoE are predicted periplasmic Cu chaperones [25,272,273], which are also linked to Cu resistance. CopC (a homologue of PcoC) is an 11 kDa periplasmic protein that is proposed to act as a Cu carrier in the oxidizing environment of the periplasm. CopC binds both Cu(I) and Cu(II) with high affinities at two different sites to form an air-stable metal–protein complex [283,284,285]. CopC of *Pseudomonas syringae* binds Cu(I) through His and Met residues, while the Cu(II) site has a mononuclear tetragonal structure supplied by the side chains of two histidine residues [285]. However, an extensive bioinformatics study indicated that the majority of CopC proteins contains only the Cu(II) site, while only about 10% of the known sequences contain the Cu(I) site. PcoE is another periplasmic Cu-binding protein that is part of the Pco system, but unlike PcoC, PcoE has no homologue in the Cop system [275]. PcoE production is under the control of the CusRS system and strongly induced by Cu [286,287]. However, its overexpression has little effect on Cu resistance in *E. coli* [275], although its presence reduces the time required for *E. coli* to recover from Cu stress [286]. It was suggested that PcoE might provide an initial Cu sequestration role in the periplasm until the Pco system is fully induced, minimizing Cu stress [275]. PcoE shares 48% sequence identity with SilE [288], which is a *Salmonella* sp. silver (Ag) resistance determinant [289]. Both proteins contain multiple conserved metal-binding sites, implying their ability to bind metals. Apo-PcoE binds Cu and Ag non-cooperatively with the highest affinity being for Cu(I). These properties are consistent with the role of PcoE as a possible Cu sponge acting as the first line of defense against Cu stress, until the induction of the Pco system [288].

### 5.4. Regulation of Cu Export

Bacterial cells usually execute a tight regulation of Cu export through various mechanisms, primarily via transcriptional and post-transcriptional regulation, and also via allosteric mechanisms and targeted proteolysis. The transcription of genes or operons involved in Cu export are generally activated when excess Cu is detected in the cytoplasm or periplasm. Transcriptional regulators of the MerR, ArsR, or CsoR families [6,252,290] sense Cu in the cytoplasm, while two-component systems such as CusRS or CopRS monitor the periplasmic Cu pool. The periplasmic and cytosolic Cu sensing systems can be independently activated, which probably allows for the prevention of Cu toxicity in the respective compartments [252]. In eukaryotes, additional levels of regulation occur mainly by post-translational modifications and by changes in intracellular localizations [252]. We will only outline the regulatory mechanisms that govern Cu export; a more detailed coverage can be found in excellent review articles [10,252,274,291]

Metal-sensing transcriptional regulators are typically homo-oligomeric (dimeric or tetrameric) DNA-binding proteins that undergo an allosteric transition when bound to their cognate metal [292]. They usually follow a common organization in which the N-terminal DNA-binding domain (DBD) is followed by a C-terminal metal-binding domain (MBD). So far, four Cu(I)-sensing transcriptional regulator families have been identified that regulate Cu efflux [1]. MerR-like regulators (e.g., CueR) act as transcriptional activators in response to Cu, while the CsoR-, CopY- and ArsR-like regulators act as repressors [290,291], controlling access and activity of RNA-polymerase.

The MerR-like Cu sensor CueR binds Cu(I) via two conserved cysteine residues in the C-terminal MBD [185]. The apo-CueR interacts with DNA and prevents RNA polymerase from binding [293,294,295]. Upon Cu(I) binding, CueR undergoes a conformational change and up-regulates the transcription of *copA, cueO* and *copZ* via a conserved DNA-distortion mechanism [293,294,295]. The CueR levels are furthermore controlled by ATP-dependent proteolysis [296].

CsoR-like Cu sensing tetrameric repressors are widely distributed in bacteria, and in particular found in species lacking CueR- and CopY-like transcriptional regulators [297,298,299]. CsoR binds one Cu(I) per protomer in a trigonal planar geometry via two cysteine and one histidine residues. Conformational changes that occur at the N-terminus of CsoR upon Cu(I) binding likely lead to CsoR dissociation from the DNA [298]. CsoR has been mainly characterized in *M. tuberculosis* [297,298,299]. It regulates the *cso* operon encoding a Cu-exporting P_1B_-ATPase in this species as well as the expression of *B. subtilis copA* [300].

The CopY-like repressors are found exclusively in Gram-positive bacteria and are characterized by a typical helix-turn-helix-motif (HTH) [301,302]. Cu(I) is usually bound to CopY via a C-terminal CxCxxxCxC motif. In *E. hirae*, when copper availability is low, CopY preferentially binds Zn(II), dimerizes and binds specifically to the promoter of *copYZAB* operon [303]. At high copper concentrations, CopY binds two Cu(I), and the associated allosteric transition leads to its dissociation from the promoter [304]. However, it is not entirely clear how Cu(I) and Zn(I) selectively influence DNA binding. Cu loading on CopY was shown to occur in exchange with Cu(I)-loaded CopZ, an interaction that has not yet been demonstrated for other Cu(I)-sensing transcriptional regulators [165,305].

BmxR is the only known Cu(I)-sensing transcriptional regulator of the ubiquitous ArsR family [306,307,308]. Similar to CopY, BmxR binds two Cu(I) via four cysteine residues [309] and Cu(I) binding allosterically inhibits DNA binding of the BmxR dimer, allowing CopA expression.

The best-known examples of a two-component system that monitor the periplasmic Cu pool, are the CusRS/PcoRS systems that regulate the Cus- and Pco-systems in *E.coli* (see Section 5.2 and Section 5.3). The membrane-bound sensor kinase CusS senses the periplasmic Cu(I) concentration, and upon Cu(I)-induced dimerization undergoes auto-phosphorylation [257,293,310]. Transfer of the phosphoryl group to the cognate response regulator CusR subsequently activates transcription of the *CusCFBA* and *CusRS* operons [141].

Intriguingly, CueR- and CopY-like regulators are absent in *Neisseria gonorrhoeae* and *copA* expression is controlled by the MerR-like transcriptional regulator NmlR that also regulates the response to nitrosative and carbonyl stress [311,312]. Thus, in some bacteria the Cu response mechanisms are interlinked with other stress response pathways.

## 6. Periplasmic Copper Chaperones and their Targets

Periplasmic Cu chaperones are particularly important, because not only do they scavenge Cu for decreasing Cu toxicity, but they also provide Cu for the biogenesis of membrane-bound and periplasmic cuproenzymes [313,314]. This is in line with the larger variety of periplasmic Cu chaperones in comparison to cytoplasmic Cu chaperones (Figure 10).

### 6.1. CusF

CusF is encoded by the *cusFCBA* cluster (see Section 5.2) and its transcription is strongly induced at high Cu concentrations [256]. Its 3D structure, determined by NMR, revealed a five-stranded β-barrel, resembling an oligonucleotide-binding-fold, which is rather untypical for Cu-binding proteins [315] (Figure 9). The metal-binding site of CusF consists of one histidine and two methionine residues, facing towards the inside of the β-barrel. An additional tryptophan residue is also involved in ligating Cu(I) or Ag(I) [191,315,316], although its exact function is not entirely clear, since its replacement with methionine increases the CuF affinity to Cu(I). It has been shown that the tryptophan residues in the extracellular domain of the eukaryotic Cu transporter Ctr4 protect the bound Cu(I) against oxidation [317], which might also be important for CusF in the oxidizing periplasmic environment.

CusF interacts with the periplasmic docking protein CusB of the CusCBA complex [256,270], but only about 40% of the periplasmic CusF participates in this reaction [256], while the remainder of it could be interacting with CopA [105]. Available data indicate that CusF is primarily required for Cu extrusion, and not involved in cuproprotein biogenesis.

### 6.2. CopI

CopI is a periplasmic plastocyanin-like protein that is induced under Cu stress [19]. *CopI* was initially discovered in the purple bacterium *Rubrivivax gelatinosus* in a cluster of genes which also contained *copA* [318]. In *R. gelatinosus*, a mutant lacking *copI* is more sensitive to Cu than one lacking *copA*. CopI contains a conserved Cu binding motif and binds Cu, but the molecular and biochemical mechanisms of Cu binding are still unknown [19]. Recently, Cd(II) was shown to increase CopI production, suggesting a possible role of this protein in general metal stress [20].

### 6.3. Periplasmic Cu Chaperones for Cox Assembly: Sco1, PCu_A_C and Cox11

Sco-like chaperones (SenC, PrrC) (Figure 10) were first identified as putative assembly factors for *aa*_3_-Cox in yeast [319] and shown to be required for the metalation of the surface-exposed binuclear Cu_A_-center of its subunit 2 [27,34,320]. Sco-like chaperones are widely distributed in eukaryotes and prokaryotes [321], but also present in bacteria lacking *aa*_3_-Cox [46,322]. Thus, they may be involved in other roles, beyond *aa*_3_-Cox assembly. Indeed, their involvements in oxidative stress response [323,324] and in the assembly of the deeply membrane-buried Cu_B_-center of subunit I of *cbb*_3_-Cox have been demonstrated [46,47,52,255]. This finding is in line with the concept that Cu insertion into membrane-bound cuproproteins generally occurs from the periplasmic side. Sco-like chaperones can bind both a single Cu(I) or Cu(II) ion with similar affinities [47,325], and they are tethered to the bacterial cytoplasmic membrane by a single TM, with a globular domain exposed to the periplasm. Cu is coordinated in a trigonal geometry via two cysteine residues of a conserved CxxxC motif and a conserved distant histidine residue. Their 3D structures have been obtained from several sources, and reveal a thioredoxin-like fold with a central four-stranded β-sheet that is flanked by three α-helices [326,327]. A solvent-exposed β-hairpin loop, which contains the histidine ligand, connects one of the α-helices with the β-core and shows the largest conformational change between the apo-Sco and Cu-bound structures. This loop is locally disordered in the apo-form, but it transitions into a well-defined structure upon metal binding [328]. This feature is typical for Sco-like chaperones, but absent in thioredoxins [328]. It is likely that this transition modulates the interaction of Sco proteins with their partner proteins, such as *cbb*_3_-Cox [46] or PCu_A_C-like chaperones [47,52]. The absence of Sco-homologues in different bacteria prevents *aa*_3_-Cox and *cbb*_3_-Cox assembly [46,48,53], without affecting Cu sensitivity, suggesting that their primary role is linked to Cox biogenesis and not to Cu toxicity. Cox deficiency in the absence of Sco proteins can be restored by increasing the Cu concentration in the media, indicating that they are particularly required at limited Cu availability [46]. In *R. capsulatus*, Cu-dependent rescue of *cbb*_3_-Cox activity is not observed in the absence of the CopA2-type ATPase CcoI [31,172], suggesting that Cu has to be routed through the cytoplasm before it binds to the ScoI-homologue SenC. A possible link between CcoI and SenC is also suggested by the transcriptional upregulation of *senC* at high Cu concentrations when CcoI is absent [46].

In eukaryotes, ScoI-like chaperones receive Cu from the small soluble Cu chaperone Cox17 [319,329], which is absent in prokaryotes [320]. Instead, bacterial ScoI-like chaperones interact with PCu_A_C-like chaperones (*periplasmic Cu_A_ chaperone*), which are soluble periplasmic Cu-binding chaperones, exclusively found in bacteria [51,328] (Figure 10). PCu_A_C-like chaperones were initially linked to the assembly of the Cu_A_-center of *aa*_3_-Cox, in cooperation with Sco1 [51]. However, further data demonstrated that the PCu_A_C-homologue of *R. capsulatus*, PccA (*Periplasmic copper chaperone A*), is also required for the assembly of Cu_B_-center of *cbb*_3_-Cox [46,47,52]. PCu_A_C-like chaperones contain a conserved H(M1)x_10_M2 × _21_HxM3 sequence motif present in a β-barrel-folded globular domain. They bind a single Cu(I) ion via the surface exposed Met3-His ligands with picomolar affinity [47,52], but can bind a second Cu(II) via the unstructured C-terminus [27]. A direct interaction between PccA and SenC was seen in *R. capsulatus*, and Cu transfer between these proteins was established in vitro, suggesting that PccA might have a Cox17-like function in *R. capsulatus* [47,52]. Although the cooperation between ScoI-like and PCu_A_C-like chaperones has been demonstrated in several species, some variations exist regarding the function of these two proteins. At low Cu availability, a Δ*pccA* strain of *R. capsulatus* still assembles small amounts of *cbb*_3_-Cox, whereas basically no *cbb*_3_-Cox is detectable in the Δ*senC* strain [52]. Similar observations were also made in *R. sphaeroides* and *B. japonicum* [48,53,255]. Thus, SenC can partially suffice for *cbb*_3_-Cox assembly even in the absence of PccA. In vitro studies using purified SenC and PccA demonstrated that Cu transfer between these proteins could occur in both directions, although transfer from PccA to SenC is more efficient. This is in line with their comparable Cu affinity (Table 3) [47]. In contrast, in vitro studies on the assembly of *ba*_3_-Cox in *T. thermophilus* indicated that PCu_A_C can insert Cu into the reduced subunit II even in the absence of Sco1 [51]. It was therefore proposed that Sco1 acts as an oxidoreductase, rather than a Cu(I) donor. In this model, Sco1 would reduce the cysteine residues in subunit II prior to Cu(I) delivery by PCu_A_C. A more recent analysis on Cu_A_ center assembly of *aa*_3_-Cox in *B. japonicum* has provided exciting mechanistic details about this assembly process [27]. First, the membrane-bound thioredoxin-like protein TlpA reduces the Cu-binding cysteine residues of subunit II. The reduced subunit II then forms a stable complex with Cu-loaded Sco1. This complex is subsequently dissociated by PCu_A_C, which is loaded with one Cu(I) in its globular domain and a second Cu(II) in its C-terminal extension. As only the C-terminal extension of PCu_A_C can transfer Cu(II), a second copy of Cu-loaded PCu_A_C is required for the formation of the binuclear Cu_A_ center [27,49]. Thus, the available data indicate that although PCu_A_C and Sco1 are conserved chaperones for Cox assembly in bacteria, they act differently on the surface exposed Cu_A_ centers of *aa*_3_-Cox and *ba*_3_-Cox and also differently on the membrane-buried Cu_B_ center in *cbb*_3_-Cox.

Similar to Sco-like chaperones that are required for Cu_B_ center assembly in *cbb*_3_-Cox, the membrane bound periplasmic Cox11 chaperone is required for the insertion of Cu_B_ into subunit I of *aa*_3_-Cox (but not *cbb*_3_-Cox) of α-proteobacteria and mitochondria [320,330] (Figure 10). Cox11 was first identified as an essential factor for *aa*_3_-Cox assembly in yeast mitochondria [331]. A Δ*cox11* strain of *R. sphaeroides* assembles *aa*_3_-Cox lacking Cu_B_ [330]. Cox11 is anchored in the membrane by a single transmembrane helix, while the soluble domain containing the Cu-binding residues is located in the intermembrane space of mitochondria or the periplasmic space of bacteria [332,333]. The solution structure of the soluble domain of Cox11 was obtained from *Sinorhizobium meliloti* showing an immunoglobulin-like fold with two highly conserved cysteine residues as Cu binding motif located on one side of the β-barrel structure [334].

### 6.4. Copper Chaperones for Nitrous Oxide Reductase (NosZ): NosL, SenC2 and PCu_A_C

Nitrous oxide reductase (NosZ) catalyzes the two-electron reduction of N_2_O to N_2_ [335,336,337]. NosZ is a periplasmic protein exported to the periplasm via the TAT-pathway [29]. NosZ forms a tight head-to-tail homodimeric metalloprotein in which the Cu_A_ site of one protomer is in close proximity to the Cu_Z_ site of the other protomer forming a composite active site at the dimer interface [84] (Figure 10). The Cu_A_ center resembles the Cu_A_ site of Cox and contains two Cu ions that are bridged by two conserved cysteine residues and further coordinated by methionine, histidine and tryptophan ligands. The Cu_Z_ center contains four Cu atoms coordinated by seven conserved histidine residues and by one or two thiols, depending on the presence or absence of O_2_, respectively [335].

The assembly process of NosZ occurs largely in the periplasm [338,339,340], and the formation of the Cu_Z_ center requires co-expression of a multi-protein assembly apparatus encoded by the *nosDFYL* operon. The *nosDFYL* operon is located downstream of *nosZ* and present in the genomes of all denitrifying species sequenced so far [341]. NosL is a monomeric protein of 18 kDa that specifically and stoichiometrically binds one Cu(I) ion [44]. Its N-terminal sequence contains a periplasmic Sec export signal and a lipobox that is required for membrane-anchorage via a tri-acylated N-terminal Cys residue [342]. The three-dimensional solution structure of apo-NosL shows two largely independent homologous domains that adopt an unusual ββαβ topology [341]. A ∆*nosL* mutant shows no defect in N_2_O reduction under Cu-sufficient conditions but fails to reduce N_2_O under Cu-limited conditions. In NosZ isolated from ∆*nosL* cells under Cu-limiting conditions, the Cu_A_ center was unaffected, while the Cu_Z_ center was impaired [45]. The formation of the Cu_Z_ center also requires the ABC transporter NosFY and the accessory protein NosD [40]. The Cu_A_ site of NosZ is less complex and likely differently assembled than the Cu_Z_ site. Deleting *pcuC* (a PCu_A_C homologue), *senC2* (a ScoI homologue) and Pden_*4445* (a protein of unknown function) of *P. denitrificans* significantly decrease N_2_O reduction under low Cu concentrations [343]. Increasing the Cu concentration could restore the phenotype of *Pden_4445* or *senC2* mutants, but not that of the *pcuC* mutant. Thus, the assembly of the Cu_A_ center of NosZ is more closely related to that of the Cu_A_ center of Cox (Figure 10), whereas a NosL-dependent assembly pathway is used for the Cu_Z_ center.

### 6.5. Copper Chaperones for Laccase-Like Multi-Copper Oxidases (MCO): CopG

Laccase-like MCOs (also called CutO, PcoA or CueO) are members of the widely distributed MCO family with unique structural, spectroscopic and functional properties. Members of this family include ceruloplasmins (ferroxidases), ascorbate oxidases, laccases, laccase-like MCOs and nitrite reductases [344]. MCOs are capable of oxidizing a broad range of phenols, aromatic amines and other chemical compounds with the concomitant reduction of molecular oxygen to water. Laccase-like MCOs are typically monomeric proteins that consist of three cupredoxin domains, which are structurally present in two ß-sheets arranged in a beta-barrel shape [36]. Three different Cu centers, named T1, T2 and T3, are involved in the redox reactions catalyzed by MCOs [345] (Figure 10). In several genomes, the *cutO*/*cueO* genes are clustered together with two additional genes, *copG*/*cutG* and a largely uncharacterized gene, termed *cutF (rcc02111)* in *R. capsulatus*.

Periplasmic CopG-like proteins are widely distributed in Gram-negative bacteria and also found genetically linked to the *copA* and *cusCBA* Cu export systems. They contribute to Cu resistance under anaerobic conditions and consist of a thioredoxin-like domain that binds a tetranuclear Cu cluster via a conserved CxCC motif and two adjacent methionine residues [346] (Figure 10). It has been proposed that CopG oxidizes Cu(I) to Cu(II), and also transfers Cu to other cuproproteins, such as CueO [346]. In *R. capsulatus*, *cutG* (*rcc02109*) is part of the *cutFOG* operon [254]. It was shown that the transcription of this operon strictly depends on the promoter located upstream of *cutF* and that the *cutF*-*cutO* intergenic region is essential for Cu dependent expression of *cutO* and *cutG* [293,294]. Comparative differential cuproproteome analyses of *R. capsulatus* revealed that the CutO and CutG protein levels are greatly increased in the presence of Cu [74].

In contrast to *cutG*, *cutF-like* genes are found only in a small subset of bacteria. In *R. capsulatus*, *cutF* encodes a putative 118 amino acid long protein with a typical Sec signal sequence and is involved in Cu resistance. Sequence comparison of different *cutF* homologues revealed the presence of a putative Cu binding CxxxC motif in the middle of the protein, and a conserved PPR sequence motif of unknown function at the C-terminus. In contrast to the expression of *cutO* and *cutG*, which were induced upon Cu supplementation, no significant increase in CutF was observed by using either promotor fusions or label-free proteomics [74,293,294]. These results suggest that only low levels of CutF are required to maintain its function for Cu tolerance. Deleting *cutF* increases Cu sensitivity and reduces drastically CutO activity, whereas deleting *cutG* has only a small effect on CutO activity (Öztürk et al., unpublished). These results possibly indicate that CutF acts as a Cu chaperon for CutO assembly, while CutG and CutO might have partially overlapping functions.

### 6.6. CueP

CueP was initially identified as a Cu resistance gene in *Salmonella enterica serovar typhimurium*, which has evolved to survive in the phagosomes of macrophages [347]. CueP expression is induced by CueR and defined as a periplasmic Cu-binding protein [58,348]. The crystal structure of CueP revealed a V-shaped dimeric structure in which each protomer consists of a N-terminal (residues 22 to 79) and C-terminal domain (residues 80 to 179) [349] (Figure 10). The N-terminal domain has a mixed α/β fold consisting of two α-helices and three β-strands. The larger C-terminal domain comprises a four-stranded β-sheet and a two-stranded β-sheet that face each other. Conserved cysteine and histidine residues are clustered on the surface of one side of the C-terminal domain providing the Cu binding site. In addition to its role in periplasmic Cu homeostasis, CueP also associates with P_1B_-type ATPases CopA and GolT to provide Cu ions for the biogenesis of Cu/Zn-superoxide dismutase in *Salmonella* [348]. Mutants lacking CueP accumulate inactive Cu/Zn-superoxide dismutase in the periplasm in vivo**, and the inactive Cu/Zn-superoxide dismutase can be reactivated in vitro by incubation with Cu-loaded CueP [348,350]. Thus, CueP seems to have a dual role by transferring Cu to the active site of Cu/Zn-superoxide dismutase and by conferring protection against high Cu concentrations.

## 7. Outlook

In this review, we provided an overview of the components of an intricate protein network that governs Cu homeostasis in prokaryotes. The complexity of this system is rationalized by the need to balance the response to potentially toxic Cu with the Cu demand for the biogenesis of key cuproenzymes, such as Cox or N_2_O reductase. The response to excess Cu has been intensively studied in several bacterial model organisms and involves largely conserved strategies: (1) Cu sensing and activation of transcriptional networks; (2) increased production of Cu binding proteins and Cu efflux pumps; (3) increased production of multi-copper oxidases and similar proteins. By combining these strategies with more general mechanisms, such as maintaining the cellular redox balance, bacterial cells can essentially prevent Cu-induced toxicity. This is particularly important for pathogenic bacteria because their eukaryotic host cells use Cu poisoning as a defense mechanism [11,12]. Understanding the bacterial Cu response will likely aid the development of future Cu-based antimicrobial compounds and surface materials.

Different from the bacterial response to Cu toxicity, the mechanisms that maintain a certain Cu quota for cuproenzyme biogenesis and the pathways that insert Cu into apo-cuproenzymes are much less explored. Only in a few cases, limited knowledge about Cu insertion into cuproenzymes with different Cu containing metal centers is available and the specific properties of the cytosolic Cu pool that is used for cuproenzymes are mostly unknown. Deleting cytosolic Cu chaperones, such as CopZ, has a profound effect on Cu resistance, but this only slightly impairs Cox activity. In contrast, deleting periplasmic chaperones, such as ScoI/SenC, drastically reduces Cox activity without increasing Cu sensitivity [46,172]. Thus, the metabolic Cu supply and the Cu detoxification systems appear to operate largely independently. Nonetheless, tight connections between both systems still exist [109] but how this coordination occurs is largely unknown.

Most cuproproteins bind Cu with affinities that are almost equivalent to a covalent bond [351,352]. Although this feature is beneficial for sequestering Cu in different cellular compartments, it also complicates Cu release and Cu transfer between the proteins. To which extent these processes are influenced by protein–protein interactions, post-translational modifications and by cellular abundance of the individual proteins, requires further analyses. Moreover, the influence of oxygen availability on Cu homeostasis also remains unclear, because most studies were performed under aerobic conditions, which favors the presence of Cu(II).

Functional and structural characterizations have been achieved for many proteins of the Cu homeostasis network, but mechanistic details are still missing in some cases. This even applies to the well-studied P_1B_-type ATPases, where the exact role of the N-terminal MBS, the Cu passage from the transmembrane MBS to the periplasm, its interaction with specific periplasmic chaperones and details on the kinetic differences between the CopA1- and CopA2-types of P_1B_-type ATPases deserve further explorations.

Remarkably, many of the proteins involved in bacterial Cu homeostasis have eukaryotic homologues that are closely associated with different metabolic disorders, including Menkes and Wilson disease [353]. In addition, targeting Cu metabolism has emerged in recent years as a novel strategy in cancer treatment, and several growth factors, transcription regulators and signaling molecules were shown to respond to cellular Cu availability [354,355,356,357,358]. Thus, understanding the general and conserved principles of Cu handling in both prokaryotic and eukaryotic cells undoubtedly continues to provide uncharted avenues for developing novel treatment regimens for human diseases.

## Figures and Tables

**Figure 1 membranes-10-00242-f001:**
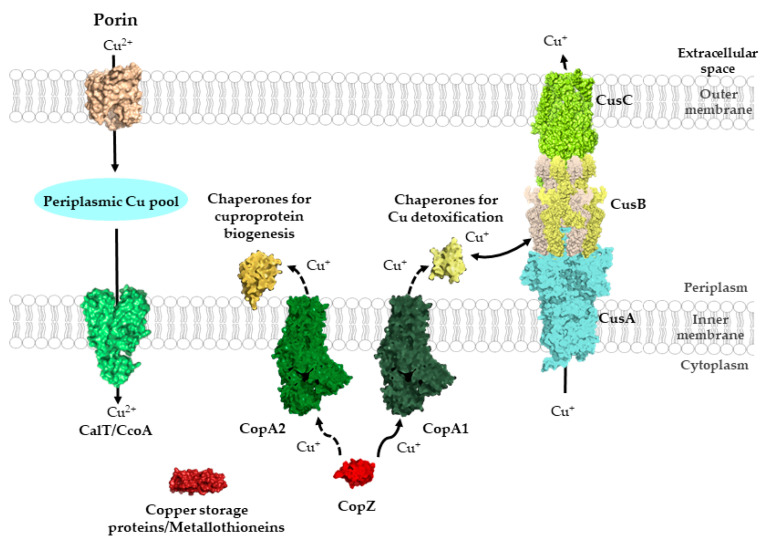
General view of Cu transport across bacterial membranes. Cu can cross the outer membrane of bacteria via porins and major-facilitator superfamily members, such as CcoA, can import periplasmic Cu into the cytosol. Additional Cu importers likely exist but have not been characterized in detail. Cytosolic Cu is bound by Cu storage proteins, metallothioneines and Cu chaperones, such as CopZ. Cu-chaperones also deliver Cu to P_1B_-type ATPases for export into the periplasm. Kinetic differences distinguish CopA1-like ATPases, which are involved in Cu detoxification, and CopA2-like ATPases, which export Cu for cuproenzyme biogenesis. CopA1-like ATPases are the primary interaction partner of CopZ, while the interaction with CopA2-like ATPases is particularly important at low Cu concentrations (dashed arrow). In the periplasm, different types of chaperones transfer Cu either for cuproenzyme biogenesis or for Cu export systems, as with the CusABC system. The structures shown were retrieved from the protein database (PDB) with the following IDs: 2ZFG (OmpF for Porin), 3WDO (YajR for CalT/CcoA), 5NQO (Csp3 for copper storage proteins), 1K0V (CopZ), 3j09 (CopA1, CopA2) 4WBR (ScoI/SenC, chaperones for cuproprotein biogenesis), 2VB2 (CusF, chaperones for Cu detoxification), 3KSS (CusA), 3H94 (CusB), 4K7R (CusC), and are depicted using Pymol.

**Figure 2 membranes-10-00242-f002:**
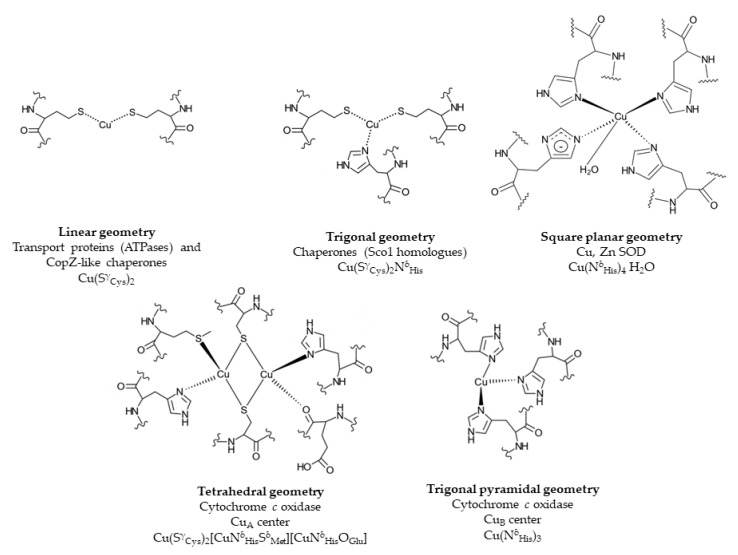
Examples of Cu binding sites in Cu trafficking proteins and cuproenzymes. Cu-transporting P_1B_-type ATPases and Cu chaperones, such as CopZ, contain a linear CxxC Cu binding motif, while in Cu chaperones, e.g., Sco1, a distal histidine residue provides an additional ligand, resulting in a trigonal planar geometry. In the cuproenzyme Cu, Zn superoxide dismutase (SOD), four histidine residues ligate Cu in a square planar geometry. The Cu_A_ center of cytochrome oxidase displays a tetrahedral geometry with two cysteines, two histidines, one glutamate and one methionine residue. In the CuB center, Cu is ligated by three histidine residues. Chemical structures were generated using ChemDoodle. The figure was adapted and modified from [54].

**Figure 4 membranes-10-00242-f004:**
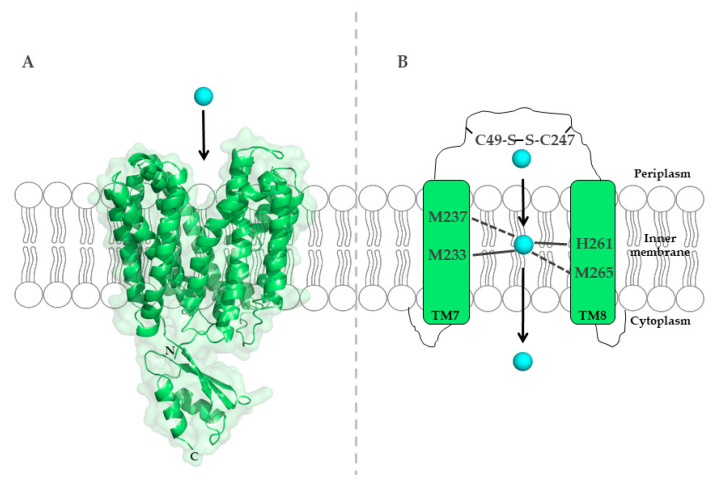
The major-facilitator-superfamily (MFS)-like Cu importer CcoA from *R. capsulatus*. (**A**) Structure of *E. coli* YajR (PDB 3WDO), which is homologous to *R. capsulatus* CcoA, but contains a large cytosolic domain, which is absent in CcoA. (**B**) Schematic representation of the metal-binding site in *R. capsulatus* CcoA, which is composed of methionine and histidine residues in transmembrane helices 7 and 8. Cysteine residues in the periplasmic loop likely contribute to Cu transport via CcoA.

**Figure 5 membranes-10-00242-f005:**
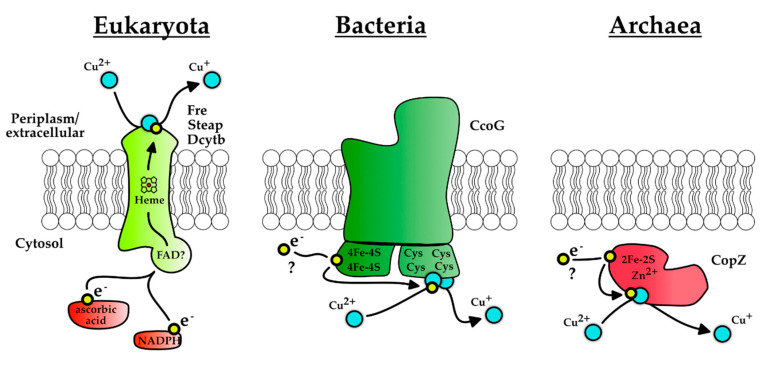
Schematic depiction of domain-specific copper-reducing enzymes. Eukaryotic cupric reductases are heme-containing, integral membrane proteins, such as Dcytb, or proteins of the Fre- and Steap-families. They contain an extracellular metal-binding site that is primarily used for Fe(III) reduction, but can also reduce Cu(II). Intracellular reductants, such as ascorbic acid or NADPH, serve as electron donors. The bacterial cupric reductase CcoG is specific for Cu(II) and reduces Cu(II) via two iron–sulfur clusters and an unknown physiological electron donor. The only described archaeal cupric reductase is a derivative of the cytosolic CopZ, using an iron–sulfur cluster and an unknown electron donor for reduction.

**Figure 6 membranes-10-00242-f006:**
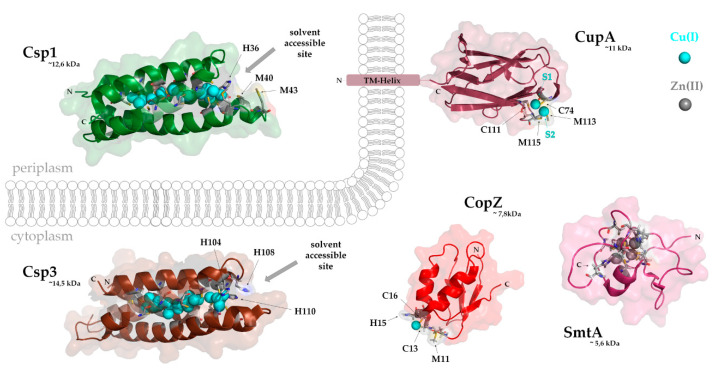
Cu chaperones and Cu storage proteins. Csp1 and Csp3 are Cu-storage proteins, which mainly differ by the presence of a cleavable Tat-signal sequence in Csp1-type proteins, which are translocated into the periplasm. CupA is a membrane-anchored Cu chaperone that has been identified in several *Streptococcus* species and which contains two Cu binding sites, S1 and S2. It has been suggested that CupA has also cupric reductase activity. CopZ is the prototype of a cytosolic Cu chaperone and homologous are present in all domains of life. CopZ-like proteins contain a single Cu binding site. SmtA is an example of a bacterial metallothionein and was isolated from *Synecococcus* sp. SmtA binds multiple Cu or Zn ions via cysteine residue and its structure was solved as Zn-SmtA. Structures were retrieved from the protein database (PDB) with the following IDs: 5FJE (Csp1), 5NQO (Csp3), 4f2e (CupA), 1K0V (CopZ), 1JJD (SmtA), and depicted using Pymol. Cu-binding residues are indicated, and Cu ions are shown as cyan spheres. For details, see the text.

**Figure 8 membranes-10-00242-f008:**
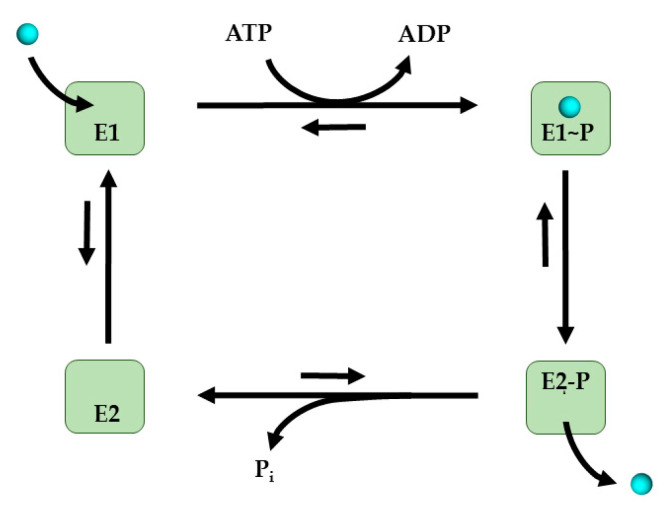
The Post-Albers cycle of P-type ATPases. For details, see text. Cu is shown as cyan sphere.

**Figure 9 membranes-10-00242-f009:**
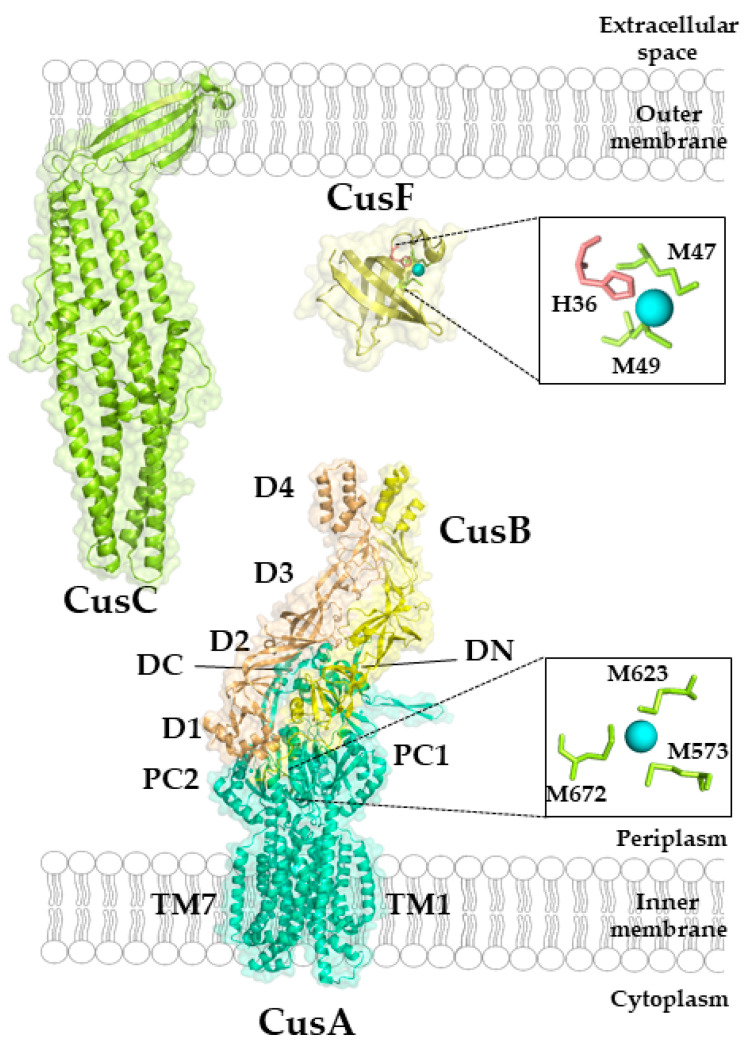
The CusABC system for Cu export. CusA is an integral membrane protein that binds two CusB copies to form a CusA_3_-CusB_6_ complex. The methionine triad for Cu/Ag binding in CusA is depicted in the insert. CusC is the outer membrane component that forms a trimer in the outer membrane and interacts with CusB. However, the complete structure of the CusA_3_B_6_C_3_ complex has not been solved yet. The CusA_3_B_6_C_3_ complex can transport Cu from the cytoplasm into the extracellular space but can also receive Cu from the periplasmic Cu chaperone CusF, which binds Cu via methionine and histidine residues (insert). Structures were retrieved from the protein database (PDB) with the following IDs: 3KSS (CusA), 3NE5 (CusBA), 4K7R (CusC), 2VB2 (CusF), and depicted using Pymol.

**Figure 10 membranes-10-00242-f010:**
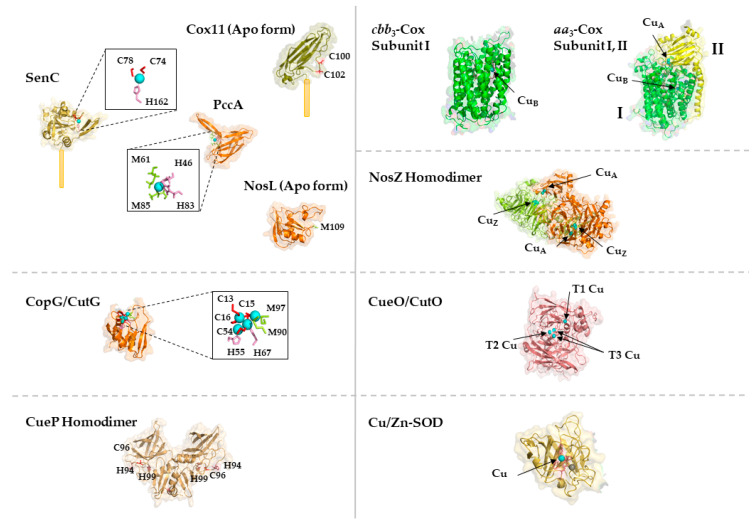
Periplasmic Cu chaperones and their target cuproproteins. The structures of Cu-chaperones are shown in the left and their corresponding target proteins in the right panels. The respective Cu binding sites are indicated, and Cu ions are shown in cyan. The ScoI-homologue SenC and Cox11 contain a single transmembrane domain of unknown structure indicated by a yellow box. NosZ, which was crystallized as homodimer, contains two Cu binding sites; for the Cu_A_ center, PccA/PCu_A_C and SenC/Sco1 are required, while NosL is involved in the formation of the Cu_Z_ center. Structures were retrieved from the protein database (PDB) with the following IDs: 4WBR (ScoI/SenC), 2K70 (PCu_A_C/PccA), 1SP0 (Cox11), 2HQ3 (NosL), 6WIS (CopG), 4GQZ (CueP), 5DJQ (*cbb*_3_-Cox), 3FYE (*aa*_3_-Cox), 6RZK (NosZ), 3OD3 (CueO), 1EQW (Cu/Zn SOD), and depicted using Pymol.

**Table 1 membranes-10-00242-t001:** Examples of bacterial cuproenzymes. Listed are cuproenzymes that stably bind copper in their catalytic center. Cuproproteins, which only transiently bind copper, such as Cu-chaperones, Cu-binding proteins or Cu-responsive transcriptional regulators, are described in the main text.

Protein	Localization	Function	Reference
Plastocyanin	Thylakoid lumen	Photosynthetic electron transfer	[26]
Cytochrome *c* Oxidase	Membrane	O_2_-reduction	[34]
Particulate Methane monooxygenase	Membrane- associated	Methane hydroxylation	[35]
Multi-Copper oxidases	Periplasm	Cu detoxification	[36]
Nitrite reductase	Periplasm	Nitrite reduction	[37]
Azurin	Periplasm	Respiratory electron transfer	[38]
Cu-Zn Superoxide dismutase	Periplasm	Superoxide detoxification	[39]
Nitrous oxide reductase	Periplasm	Denitrification	[40]
Amine oxidases	Periplasm	Amine oxidation	[41]
MccA-type Sulfite reductase	Periplasm	Sulfite reduction	[42]
Tyrosinase	Extracellular	Monooxygenase	[43]

**Table 2 membranes-10-00242-t002:** Known bacterial cupric reductase activities.

Protein	Organisms	Function	Reference
CcoG	*Rhodobacter capsulatus*	Specific Cu(II) reduction via two tetranuclear iron–sulfur clusters	[157]
NDH-2	*Escherichia coli*	NADH-Dehydrogenase 2, unspecific quinol-dependent Cu(II) reduction	[162]
?	*Lactococcus lactis*	Unspecific quinol-mediated Cu(II) reduction	[163]
?	*Pseudomonas* sp.	Cell-free Cu(II) reduction, not further characterized	[164]

**Table 4 membranes-10-00242-t004:** Classification of P_1B_-ATPases.

P_1B_-ATPase Subgroup	Substrate	References
P_1B_-1	Cu(I), Ag(I)	[112,221,222]
P_1B_-2	Zn(II), Cd(II), Pb(II)	[221,222]
[P_1B_-3]	Reclassified as P_1B_-1	[220]
P_1B_-4	Co(II), Cd(II), Zn(II)	[223]
P_1B_-5	Ni(II), Fe(II)	[224]
P_1B_-6	Fe(II)	[222]
P_1B_-7	Unknown	[222]

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
