# Peer review of "Cu Homeostasis in Bacteria: The Ins and Outs"

_membranes, 2020, doi:10.3390/membranes10090242_

Round 1

Reviewer 1 Report

The authors took a big mission to describe copper mechanism in all bacteria species. They wrote a very long and detailed review which is remarkable, however I thinks that this is too long, very hard to follow and the final prospective of the all review is missing.

Therefore, I believe that extensive rewriting and also English editing is required. Beside that there are several misleading information. Not all mechanisms exist in all bacteria species, this should be better explained and clarified. There is a difference between gram negative and gram positive bacteria. There are also mis-correct information for example: CusCBA efflux only Cu(I) and not Cu(II). the mechanism described CueR and CopZ is not complete, and probably there are more missing and incomplete data since this paper deals with too many proteins and systems, so a reliable picture on all proteins is very hard to provide. Maybe it is better to delete some of the systems described here and to focus on systems that these authors are better familiar with. 

Author Response

We thank the reviewer for his comments and suggestions.

  1. We agree with the reviewer that our review article intends to provide a broad overview on Cu trafficking mechanisms in bacteria. As such, the article covers a lot of information and we have re-organized the manuscript and also edited the text to make the manuscript more comprehensible.
  2. We have now also more clearly differentiated between Gram-positive and Gram-negative bacteria.
  3. We have also corrected the error in Figure 1 regarding Cu(II) efflux by the CusCBA system.
  4. We also corrected the error regarding the CueR/CopZ affinities.
  5. After careful reading of our manuscript, we did not identify any further errors.
  6. The aim of this review is to provide a comprehensive overview of the mechanisms involved in Cu import/export and trafficking in the cells. We choose to address different mechanisms and systems (rather than specific ones) to cover this issue and to provide information that will be of interest to a large scientific community. Since the review tackles different issues, we choose to emphasize the most significant information and data on the described systems (properly referenced in the text as indicated by all three reviewers) and less significant information might be therefore missing. In any case, the review provide the readers with information about most of the known mechanisms with special emphasis on the mechanisms that the authors are working/familiar with (like CcoA, the Cu chaperones, CopA/CcoI, CcoG). Deleting some parts would also not be in line with the evaluation of the two other reviewers, who referred to our article as excellent and very thorough.

Reviewer 2 Report

This is an excellent review on bacterial Cu homeostasis and associated proteins. The authors should consider including the following minor comments

  • Consider including schematic representation of Cu binding sites to support statements on line 105-113, and to further provide visual background to figure 3.
  • Revise for reference format and duplication i.e. 80 and 151; 1 and 133; 201 and 260… and others
  • Please consider: https://doi.org/10.1186/1472-6807-12-25 , DOI: 1007/s10534-015-9860-x , https://doi.org/10.1186/s40659-020-00274-7 , DOI:10.1128/JB.01847-07, https://doi.org/10.1073/pnas.1009261108  , doi:10.1039/c4mt00305e.

A brief description of Cu management and virulence of bacteria such as tuberculosis would be a nice addition.

Author Response

This is an excellent review on bacterial Cu homeostasis and associated proteins. The authors should consider including the following minor comments

Response:

We thank the reviewer for the positive evaluation of our article.

  • Consider including schematic representation of Cu binding sites to support statements on line 105-113, and to further provide visual background to figure 3.

Response:

This is an excellent suggestion and we have included a new figure (Figure 2) in the manuscript.

  • Revise for reference format and duplication i.e. 80 and 151; 1 and 133; 201 and 260… and others

Response:

We thank the reviewer for identification of duplicates in the reference list, which we have removed.

  • Please consider: https://doi.org/10.1186/1472-6807-12-25 , DOI: 1007/s10534-015-9860-x , https://doi.org/10.1186/s40659-020-00274-7 , DOI:10.1128/JB.01847-07, https://doi.org/10.1073/pnas.1009261108  , doi:10.1039/c4mt00305e.

A brief description of Cu management and virulence of bacteria such as tuberculosis would be a nice addition.

Response:

We have added additional references concerning Cu management in pathogenic bacteria. However, in light of the already rather long article, we think that this important aspect about Cu homeostasis is better covered in recent review articles that we referred to in our article (lanes 42-49).

Reviewer 3 Report

This is an excellent, well written, and very thorough review of the current state of play of the various known components of bacterial copper transport and homeostasis. As such, it will be a valuable, up-to-date resource for those in this research community, and I would recommend it as essential reading for new members of my lab. I have a number of recommendations and suggestions that I feel will further improve the manuscript, detailed below in sequential order.

  1. In Figure 1, are the structures shown in the illustration (all) based/modelled on crystal structures? If so, these should be described in the legend, modelling/structure viewing software described, and appropriate references cited.

  1. I think it is important to delineate the difference between cuproproteins and cuproenzymes throughout the review. Currently, the term ‘cuproproteins’ is used to describe all copper binding proteins, but I feel it would be more valuable to define the difference between proteins and enzymes. The latter are exclusively ‘end-users’ of copper, whereas the former (perhaps with the exception of plastocyanin, although some would argue even this catalyses the transfer of electrons) are primarily copper-handling proteins, including the homeostatic proteins themselves. For example, on line 74, where ‘cuproproteins’ show a catalytic activity, they should be defined as ‘cuproenzymes’. Likewise, the legend to Table 1 could define the enzymes as stably binding copper vs. the proteins that transiently bind it.

  1. In section 2, entitled “Copper import across the outer and inner membranes in bacteria”, all but the last line is dedicated to eukaryotes. I would suggest this paragraph be amended by adding that (i) Ctr homologues are not found in (any?) bacterial genomes, and (ii) that P-type ATPases are conserved in (almost all?) bacterial genomes, to precede the subsequent sections on bacterial copper transporters.

  1. Lines 120-2, the authors state: “In eukaryotes, the mechanisms of Cu import are well known and facilitated by mainly two major families of membrane-integral proteins: the Ctr (copper transport) family of transporters and P1B-type ATPases (also known as CPx-ATPases)”. The P-type ATPases all export copper from the cytosol (including CtaA – see below). None have been shown to import. I suggest changing the word ‘import’ to ‘transport’.

  1. In Figure 2, I am uncomfortable with CtaA being shown as a possible copper importer. This was based on phenotypes of the ctaA mutant in cyanobacteria, but the Robinson group who first showed this went on to accept that CtaA effluxes copper from the cytosol. Its phenotype that implied import (as with other such ATPases, CopB) are likely due to a complex role in copper supply to extra-cytoplasmic copper enzymes. This is particularly true in cyanobacteria, which have complex intercellular membrane compartments, so CtaA may be involved in transport into those rather than into the periplasm. I believe Arguello showed that CtaA functions in the same direction as all other characterised P-type ATPases. The subsequent section (lines 270-82) also needs to be amended on this point. It is worth noting that recent proteomic analyses, that have measured the abundance of membrane proteins in cyanobacteria, have made observations of the presence of these transporters which may shed light on their relative function.

  1. In section 2.2, I think it is important to discuss the possibility that some Csps handle periplasmic copper. The original discovery of Csp1 in M. trichosporium was of a sub-type that is Tat targeted – it may reside in the periplasm, or in intracellular compartments of those unusual methanotrophic bacteria (this latter point should be clarified in lines 513-515). But other bacterial genomes, such as Neisseria gonorrhoeae, also possess Tat-targeted Csps. These likely play an important role in handling and storing periplasmic copper in Gram-negative bacteria that possess them, as is illustrated in Figure 5, and is worth mentioning briefly in this section too.

  1. In section 4.2, I feel the section on CupA would be strengthened by pointing out that the presence of cupredoxin domains on CupA and CopA in S. pneumoniae represents a convergent evolution cf. the ferredoxin domains in most CopZ/CopA pairs, and that this strongly argues that the protein-protein interaction between chaperone and pump N-terminal domain is functionally important. A similar point can be made about the presence of six MBDs in eukaryotes (682-90), illustrating the importance of their function.

  1. On lines 528-34, I wonder if the difference in release rates (likely due to the difference in copper binding residues at the opening) of the Csp classes may be a reflection of the differing affinities/kinetics of copper binding of the ‘acceptor’ enzymes in the periplasm vs cytosol?

  1. In section 5.4, I would recommend mentioning that some organisms, such as Neisseria sp., appear to have no copper sensor and the expression of the copper detoxification system is expressed constitutively. Although rare, these systems are intriguing. I would also recommend delineating more clearly in this section the difference in role of periplasmic and cytosolic copper sensors, which can be independently activated, and presumably play different roles in controlling toxicity in these very different compartments of Gram negative bacteria.

Author Response

Reviewer 3:

This is an excellent, well written, and very thorough review of the current state of play of the various known components of bacterial copper transport and homeostasis. As such, it will be a valuable, up-to-date resource for those in this research community, and I would recommend it as essential reading for new members of my lab. I have a number of recommendations and suggestions that I feel will further improve the manuscript, detailed below in sequential order.

Response:

We thank the reviewer for the positive and encouraging evaluation of our article. The important issues raised by the reviewer were addressed and helped us to improve the quality of our manuscript.

  1. In Figure 1, are the structures shown in the illustration (all) based/modelled on crystal structures? If so, these should be described in the legend, modelling/structure viewing software described, and appropriate references cited.

Response:

This information was added to the legend of Figure 1.

  1. I think it is important to delineate the difference between cuproproteins and cuproenzymes throughout the review. Currently, the term ‘cuproproteins’ is used to describe all copper binding proteins, but I feel it would be more valuable to define the difference between proteins and enzymes. The latter are exclusively ‘end-users’ of copper, whereas the former (perhaps with the exception of plastocyanin, although some would argue even this catalyses the transfer of electrons) are primarily copper-handling proteins, including the homeostatic proteins themselves. For example, on line 74, where ‘cuproproteins’ show a catalytic activity, they should be defined as ‘cuproenzymes’. Likewise, the legend to Table 1 could define the enzymes as stably binding copper vs. the proteins that transiently bind it.

Response:

This is a very valid point and we have changed the nomenclature throughout the manuscript to cuproenzymes versus cuproproteins.

  1. In section 2, entitled “Copper import across the outer and inner membranes in bacteria”, all but the last line is dedicated to eukaryotes. I would suggest this paragraph be amended by adding that (i) Ctr homologues are not found in (any?) bacterial genomes, and (ii) that P-type ATPases are conserved in (almost all?) bacterial genomes, to precede the subsequent sections on bacterial copper transporters.

    Response:

    Again, this is a valid point, and we have changed the text accordingly (lanes 148-166).

  1. Lines 120-2, the authors state: “In eukaryotes, the mechanisms of Cu import are well known and facilitated by mainly two major families of membrane-integral proteins: the Ctr (copper transport) family of transporters and P1B-type ATPases (also known as CPx-ATPases)”. The P-type ATPases all export copper from the cytosol (including CtaA – see below). None have been shown to import. I suggest changing the word ‘import’ to ‘transport’.

    Response:

We agree with the reviewer and the entire part was re-phrased (lane 148-166).

  1. In Figure 2, I am uncomfortable with CtaA being shown as a possible copper importer. This was based on phenotypes of the ctaA mutant in cyanobacteria, but the Robinson group who first showed this went on to accept that CtaA effluxes copper from the cytosol. Its phenotype that implied import (as with other such ATPases, CopB) are likely due to a complex role in copper supply to extra-cytoplasmic copper enzymes. This is particularly true in cyanobacteria, which have complex intercellular membrane compartments, so CtaA may be involved in transport into those rather than into the periplasm. I believe Arguello showed that CtaA functions in the same direction as all other characterised P-type ATPases. The subsequent section (lines 270-82) also needs to be amended on this point. It is worth noting that recent proteomic analyses, that have measured the abundance of membrane proteins in cyanobacteria, have made observations of the presence of these transporters which may shed light on their relative function.

    Response:

We agree with the reviewer and have modified Figure 2 (now Figure 3). In the text, we have also addressed this information (lanes 157-162).

  1. In section 2.2, I think it is important to discuss the possibility that some Csps handle periplasmic copper. The original discovery of Csp1 in M. trichosporium was of a sub-type that is Tat targeted – it may reside in the periplasm, or in intracellular compartments of those unusual methanotrophic bacteria (this latter point should be clarified in lines 513-515). But other bacterial genomes, such as Neisseria gonorrhoeae, also possess Tat-targeted Csps. These likely play an important role in handling and storing periplasmic copper in Gram-negative bacteria that possess them, as is illustrated in Figure 5, and is worth mentioning briefly in this section too.

Response:

This is an important point and we have added this information (lanes 244-253).

  1. In section 4.2, I feel the section on CupA would be strengthened by pointing out that the presence of cupredoxin domains on CupA and CopA in S. pneumoniae represents a convergent evolution cf. the ferredoxin domains in most CopZ/CopA pairs, and that this strongly argues that the protein-protein interaction between chaperone and pump N-terminal domain is functionally important. A similar point can be made about the presence of six MBDs in eukaryotes (682-90), illustrating the importance of their function.

    Response:

    Again, this important aspect was described in more detail in the revised manuscript (lanes 516-524).

  1. On lines 528-34, I wonder if the difference in release rates (likely due to the difference in copper binding residues at the opening) of the Csp classes may be a reflection of the differing affinities/kinetics of copper binding of the ‘acceptor’ enzymes in the periplasm vs cytosol?

    Response:

    This is an interesting thought and we have added it to the article (lanes 578-580).

  1. In section 5.4, I would recommend mentioning that some organisms, such as Neisseria sp., appear to have no copper sensor and the expression of the copper detoxification system is expressed constitutively. Although rare, these systems are intriguing. I would also recommend delineating more clearly in this section the difference in role of periplasmic and cytosolic copper sensors, which can be independently activated, and presumably play different roles in controlling toxicity in these very different compartments of Gram negative bacteria.

    Response:

We agree with the reviewer and in the revised version, we have better described the difference between the periplasmic and cytosolic Cu sensors. We also included that Neisseria lacks a Cu sensor (lanes 915-968).

Round 2

Reviewer 1 Report

The authors made a good work in revising the review. I have no further comments.